# A neural network-based method for generating synthetic $1.6\,\mu m$ near-infrared satellite images

Florian Baur[1,2], Leonhard Scheck[2,1], Christina Stumpf[1], Christina Köpken-Watts[1], and Roland Potthast[1]

[1]Deutscher Wetterdienst, Offenbach, Germany
[2]Hans-Ertel-Centre for Weather Research, Ludwig-Maximilians-Universität, Munich, Germany

**Correspondence:** Florian Baur (Florian.Baur@lmu.de)

**Abstract.** In combination with observations from visible satellite channels, near-infrared channels can provide valuable additional cloud information, e.g. on cloud phase and particle sizes, which is also complementary to the information content of thermal infrared channels. Exploiting near-infrared channels for operational data assimilation and model evaluation requires a sufficiently fast and accurate forward operator. This study presents an extension to the method for fast satellite image synthesis

(MFASIS) that allows for simulating reflectances of the $1.6\,\mu m$ near-infrared channel based on a computationally efficient neural network with the same accuracy that has already been achieved for visible channels. For this purpose, it is important to better represent vertical variations of effective cloud particle radii, as well as mixed-phase clouds and molecular absorption in the idealised profiles used to train the neural network. A new approach employing a two layer model of water, ice and mixed-phase clouds is described and the relative importance of the different input parameters characterising the idealized profiles is

analyzed. A comprehensive dataset sampled from IFS forecasts together with different parameterizations of the effective water and ice particle radii is used for the development and evaluation of the method. Further evaluation uses a month of ICON-D2 hindcasts with effective radii directly determined by the two-moment microphysics scheme of the model. In all cases, the mean absolute reflectance error achieved is about 0.01 or smaller, which is an order of magnitude smaller than typical differences between reflectance observations and corresponding model values. The errors related to the imperfect training of the neural

networks present only a small contribution to the total error and evaluating the networks takes less than a microsecond per column on standard CPUs. The method is also applicable for many other visible and near-infrared channels with weak water vapour sensitivity.

## 1 Introduction

Over the last decades, satellite observations have become the most important observation type assimilated in numerical weather

prediction (NWP) systems. They dominate not only in terms of the total number of assimilated observations, but also with respect to the overall impact on the forecast quality of operational global NWP systems (Bormann et al., 2019; Eyre et al., 2022). The preferred way to assimilate satellite observations from imagers and sounders is the direct assimilation of radiances, which requires a forward operator to generate synthetic radiances from the NWP model state. In contrast to assimilating retrievals, no external a priori information (e.g. from other models or climatologies) is required in the direct assimilation

approach and in general also the characterization of errors is less problematic (Errico et al., 2007). Satellite radiances are increasingly assimilated not only in clear-sky conditions, but also in the presence of cloud and precipitation. This so called 'all-sky' approach is being applied successfully for microwave (MW) observations in different global NWP systems (Geer et al., 2018) and progress is being made towards the direct all-sky assimilation of infrared (IR) observations (Geer et al., 2018, 2019; Li et al., 2022). Similarly, satellite observations are also assimilated in many regional models (Gustafsson et al., 2018) with a particular focus on observations from geostationary imagers providing temperature, moisture and cloud information with high temporal and spatial resolution (see e.g. Otkin and Potthast 2019, Okamoto 2017).

Efforts are ongoing to improve the exploitation of satellite observations that are currently underutilized, both in terms of assimilating already operational data under all conditions and over all surfaces and in using channels that are not yet directly assimilated at all (Valmassoi et al., 2022; Hu et al., 2022). Solar satellite channels fall into the latter category, mostly because sufficiently fast and accurate forward operators are missing or have only become available recently. The development of such operators was hampered by the fact that standard radiative transfer (RT) methods for the solar spectral range (with wavelengths $\lambda < 4\mu$m) are computationally very expensive, as they require the detailed modeling of multiple scattering processes, which are much more important than in the thermal part of the spectrum ($\lambda > 4\mu$m). Moreover, 3D RT effects, i.e. effects involving horizontal photon transport, e.g. related to inclined cloud tops, cloud shadows or complex topography (see Marshak and Davis 2005 for a detailed discussion) can be important for solar channels, especially at high resolutions and for large zenith angles. For visible channels Scheck et al. (2016) developed MFASIS (method for fast satellite image synthesis), an efficient 1D RT approach based on a strong simplification of the vertical cloud structure and the use of precomputed results stored in compressed look-up tables (LUTs). This LUT-based version of MFASIS has been integrated into the RTTOV satellite forward operator package in the version v12.2 with subsequent improvements in versions v12.3 and v13.1(Saunders et al., 2018, 2020). MFASIS has been used in several model evaluation studies (Heinze et al., 2016; Stevens et al., 2020; Sakradzija et al., 2020; Geiss et al., 2021) as well as data assimilation studies (Schröttle et al., 2020; Scheck et al., 2020). The assimilation of visible SEVIRI observations at $0.6\mu$m using RTTOV-MFASIS has become operational at DWD in March 2023. An extension to MFASIS to account for the most important 3D RT effects in a computationally efficient way is available (Scheck et al., 2018) and recently a faster and more flexible version based on neural networks instead of LUTs has been developed (Scheck, 2021) and integrated into RTTOV v13.2.

The cloud information contained in visible channels is complementary to that available from thermal infrared channels. Whilst visible channels provide almost no information that could be used to determine the cloud top height or discriminate frozen from liquid clouds, they contain much more information on the cloud water or cloud ice content as they saturate only for much thicker clouds than thermal channels (Geiss et al., 2021). There is also some dependency of visible radiances on the cloud particle sizes and the surface structure of clouds.

Near-infrared (NIR) channels ($0.75 \leq \lambda \leq 4\mu$m) depend on cloud particle sizes and angles in a different way, compared to visible channels, and can thus provide additional information that could be very valuable both for model evaluation and data assimilation. The combined information of visible and near-infrared channels has been successfully used for many years to simultaneously retrieve cloud optical thickness and effective particle radii (following Nakajima and King 1990). Such obser-

vations constraining the cloud microphysics are also of special relevance for NWP models employing advanced cloud physics schemes like two-moment schemes that provide prognostic effective cloud particle sizes (see e.g. Seifert and Beheng 2006). Of particular interest is the $1.6\mu$m channel available on many satellite imagers, because at this wavelength ice absorbs radiation considerably stronger than water. In combination with a visible channel, for which absorption by both water and ice is very weak, the $1.6\mu$m channel can thus provide information that is helpful for distinguishing liquid from frozen clouds (but will not in all cases allow for a clear distinction, see e.g. Fig 4 in Coopman et al. 2019). While information on the cloud phase is also available from thermal infrared channels, using near-infrared channels in addition (Baum et al., 2000) or instead (Nagao and Suzuki, 2021) can improve the reliability of retrievals. Assimilating the $1.6\mu$m channel could thus be a promising way to reduce cloud phase errors.

MFASIS can already be applied for NIR channels and LUTs for $1.6\mu$m channels of different instruments are available as part of the RTTOV package. However, mainly due to the rather approximate treatment of mixed-phase clouds, the currently employed method is considerably less accurate for this channel. Some corrections included in RTTOV 13.1 allow for avoiding the largest errors, but the accuracy for the $1.6\mu$m channel is still lower than for visible channels. This study demonstrates how to both improve the accuracy and reduce the computational effort through using a machine learning-based approach. We will focus on generating synthetic images for the $1.6\mu$m channel of the SEVIRI instrument aboard Meteosat Second Generation (MSG) from global and regional NWP model data. Building on the neural network-based results for visible channels of Scheck (2021), suitable network input parameters to account for the more complex dependency of near-infrared radiances on the atmospheric state will be identified. Networks with these input parameters are then trained and tested on different data sets.

The rest of this study is organized as follows: Data and methods are discussed in Sect. 2, suitable network input parameters are derived in Sect 3, the training of neural networks based on these profiles is discussed in Sect. 4, the full method is evaluated using different data sets in Sect. 5 (also for other solar channels) and conclusions are given in Sect. 6.

## 2 Data and methods

### 2.1 Radiative transfer methods

#### 2.1.1 DOM

For reference calculations and the generation of neural network training data, the discrete ordinate method (DOM, see Stamnes et al. 1988) is used. We rely on the implementation of DOM in the RTTOV RT package (Saunders et al., 2018). The required input data comprise vertical profiles of the cloud water and cloud ice content including the corresponding effective particle radii, a value for the surface albedo ($A$), solar and satellite zenith angles ($\theta_0$, $\theta$), and the difference of their azimuth angles ($\Delta\phi$). DOM solves the plane-parallel radiative transfer equations and computes the resulting top-of-atmosphere reflectance. In RTTOV, the liquid cloud optical properties used in this process are based on Mie (1908) and for ice clouds the optical properties for the general habit mixture of Baum et al. (2005, 2007) are used. Aerosols are neglected in this study, but clear sky Rayleigh scattering and molecular absorption are taken into account.

### 2.1.2 MFASIS

DOM generates accurate 1D RT solutions, but is significantly too slow for operational applications like data assimilation. For this reason, the fast method MFASIS (Scheck et al., 2016) was developed and has subsequently been implemented in RTTOV (beginning with version 12.2, see Saunders et al. 2020). In MFASIS, the cloud top height and details of the vertical cloud structure are not taken into account for computing the reflectance, and still the reflectance errors with respect to DOM solution are small. These properties of the input profiles can therefore be considered to be not very important for the resulting reflectance. The complex vertical profiles from NWP runs are in MFASIS replaced by highly idealised profiles with the same total optical depths and mean effective particle radii. These idealised profiles contain a homogeneous ice cloud above a homogeneous water cloud at fixed heights embedded in a standard atmosphere. Only eight parameters are used to fully characterize the idealised radiative transfer problem: the optical depths and vertically averaged effective particle radii for the water and the ice cloud, three angles to define the sun-satellite geometry and the surface albedo. Reflectances for many combinations of the parameters are pre-computed using DOM and stored in an eight-dimensional look-up table (LUT). The latter is reduced from 8GB to 21MB using a lossy compression method. To obtain reflectances for arbitrary input profiles, it is only necessary to compute the input parameters from them and interpolate the reflectance in the LUT at the corresponding location [1]. This process takes only several microseconds and is thus orders of magnitude faster than running DOM. Both the achieved speed and accuracy are sufficient for assimilation of visible radiance observations in operational applications.

Whilst the simplification of the vertical profiles in MFASIS causes reflectance errors that are acceptable for data assimilation or model evaluation using visible channels, they remain significantly larger for the $1.6\mu$m near-infrared channel that is considered in this study for three reasons:

- The absorption in ice is considerably stronger than in water. Replacing mixed-phase clouds, which are often dominated by liquid water at the top, by an ice cloud above a water cloud causes therefore large errors.

- The $1.6\mu$m channel is slightly affected by molecular absorption (due to $CO_2$, $CH_2$ and for wider channels like the one on the SEVIRI instrument considered here also water vapour), which means that the air mass between cloud and satellite will have a stronger influence on the reflectance than for visible channels[2]. For SEVIRI also the water vapour mass will have some influence. To give an example, for a relatively high column integrated water vapour content of 50mm and solar and satellite zenith angles of $60°$ the reflectance is reduced by about 5%.

- The vertical variation of the effective particle radii is not taken into account in the simplified profiles. While this error source alone would still be acceptable (it is indeed also present for visible channels), it contributes significantly to the total error, in addition to the two other problems listed above. The problem is that the effective radii in the uppermost cloud layers, from which photons can escape after single scattering events, may be different from the effective radii at higher optical depths that contribute to the reflectance by multiple scattering processes. To approximate both the correct

---

[1]A linear interpolation is performed in the seven cloud and angle dimensions, the albedo dimension is treated differently, as discussed in Sect. 4.1

[2]Visible channels have also a slight cloud top height dependence due to Rayleigh scattering

scattering angle dependence of the reflectance, which is dominated by single scattering processes, and the correct angle-averaged reflectance, which is often dominated by multiple scattering processes, at least two different radii are required.

Preliminary solutions to account for the largest errors were introduced in the MFASIS implementation in RTTOV v13.1: Replacing ice within or below water clouds with water clouds of the same optical depth reduced the errors for mixed-phase clouds. The computation of the mean effective radius was modified to give more weight to the upper cloud layers for thick clouds. While these corrections succeeded in removing the largest errors, the mean errors are still considerably larger than for visible channels. In this study we will present a new approach, which is more accurate, faster and based on neural networks.

### 2.1.3 Neural networks and MFASIS-NN

Artificial neural networks are the most popular machine learning approach. They have the advantages that mature, easy to use implementations are available and that many CPUs and GPUs now support hardware-accelerated training and evaluation of these networks. A neural network-based version of MFASIS for visible channels, in the following referred to as MFASIS-NN, was developed by Scheck (2021). While the simplification of vertical profiles in this method is the same as in MFASIS, the LUT is replaced by a deep feed-forward neural network. The input parameters of the network correspond to the dimensions of the LUT. Reflectances for arbitrary albedo values can be computed from the three output parameters approximating reflectances for surface albedo values 0, 0.5 and 1 (see Eq. 6 in Scheck 2021 and discussion in Sect. 4.1). The study shows that networks with several 1000 parameters in 4–8 hidden layers can be trained well enough to achieve reflectance errors that are in general smaller than the ones of the LUT version. The amount of data to be generated with DOM for the training is a factor 1000 smaller than the 8GB required for the LUT-based MFASIS. Moreover, using a computationally cheap activation function and a Fortran inference code optimized for small networks, MFASIS-NN is an order of magnitude faster than the LUT-based MFASIS.

As in Scheck (2021), deep feed-forward neural networks are used in this study. Networks with 8 hidden layers and 15, 25, or 32 nodes per layer are considered. The networks are initialized with random numbers and trained with the open-source Tensorflow package (Abadi et al., 2015) using standard methods. The mini-batch gradient descent method (with a batch size of 256) and the Adam algorithm (see Chapter 8 in Goodfellow et al. 2016) with a learning rate of $2.5 \times 10^{-4}$ were utilized for this purpose. About $1.4 \times 10^{7}$ synthetic training data samples were generated by assuming random numbers for the normalized network input parameters (uniformly distributed in $[0, 1]$, which means that the unnormalized, physical variables are uniformly distributed over the ranges given by Tab. 2) and computing the corresponding reflectance with DOM. 80% of the samples were used for the training, 20% served as independent validation data. The networks were trained for 4000 epochs. During the training, the updated network weights and biases were stored only if they resulted in a reduced root mean squared error of the validation data set, an approach known as early stopping. For the evaluation or inference of networks, we employed FORNADO, an optimized Fortran code including tangent linear and adjoint versions. To reduce the computational effort, the

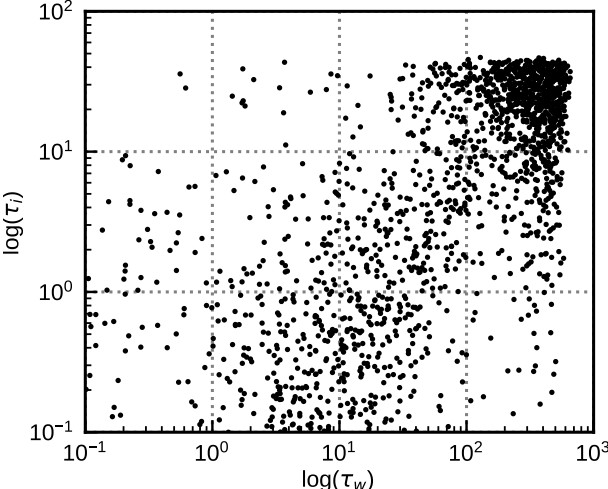

**Figure 1.** Water cloud optical depth $\tau_{\mathrm{w}}$ and ice cloud optical depth $\tau_{\mathrm{i}}$ for all profiles of the 'std' data set (see Tab. 1).

"cheap soft unit" (CSU; see Fig. 2 in Scheck 2021), defined as

$$f_{\mathrm{CSU}}(x) = \begin{cases} 0, & \text{if } x < -2 \\ -1 + 0.25\,(x+2)^2, & \text{if } x \in [-2, 0] \\ x, & \text{if } x > 0 \end{cases} \tag{1}$$

was used as an activation function for the hidden layers. This function is very similar to the well-known exponential-linear unit (ELU), $f_{\mathrm{ELU}}(x) = \min(e^x - 1, |x|)$, but does not involve a computationally expensive exponential function, which can also prevent the compiler from using vector instructions. For the output nodes, we used the softplus function, $f_{\mathrm{softplus}}(x) = \ln(1 + e^x)$, which guarantees that all output values are positive.

## 2.2 NWP-SAF profiles

A comprehensive set of profiles available from the Satellite Application Facility for Numerical Weather Prediction (NWP SAF) project is used to tune and evaluate the methods developed for this study. The data set comprises 5000 individual profiles selected from a year (1 September 2013 – 31 August 2014) of short-range forecasts produced with the Integrated Forecasting System (IFS) of the European Centre for Medium-Range Weather Forecasts (ECMWF) using an algorithm which only selects profiles that are sufficiently different in cloud variables compared to the other selected profiles. The profiles represent realistic seasonal variability and, as they are spread over the entire globe, global variability is also well represented. About $30\,\%$ of the profiles are located over land and about $40\,\%$ between the northern and southern tropics. Please refer to Eresmaa and McNally (2014) for further information about the data set. It should be noted that the cloud fraction profiles, $c(z)$, were modified for this study. To avoid having to take cloud overlap into account, for which different assumptions exist (see e.g. Scheck et al.,

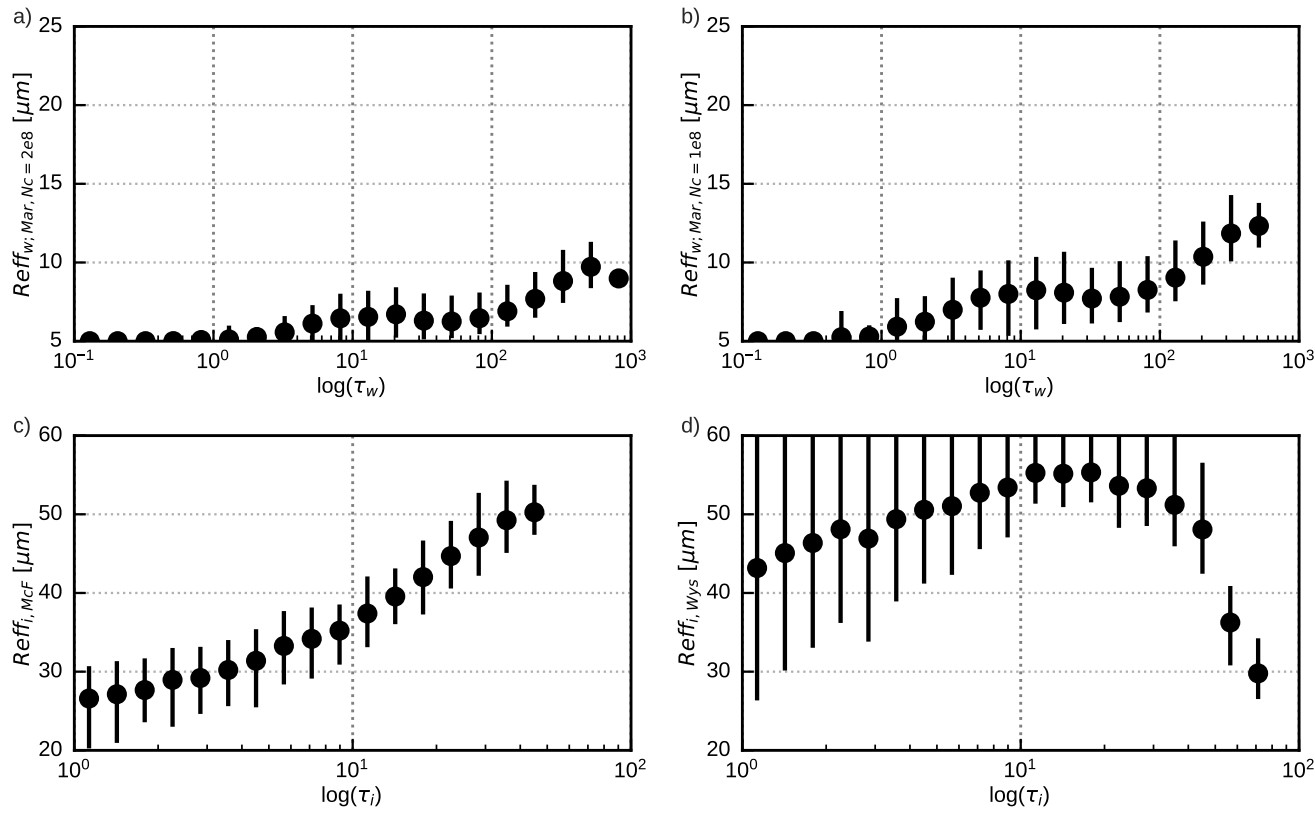

**Figure 2.** Mean vertically averaged effective particle radii (dots) in logarithmically spaced optical depth bins for the 'std' (left column) and 'rmod' (right column) profile data sets for water clouds (upper row) and ice clouds (lower row). The vertical lines connect the 5th and the 95th percentiles for each bin.

2018) and which is not in the focus of this work, the cloud fraction was set to zero for $c < \frac{1}{2}$ and to one for $c >= \frac{1}{2}$. While this simplification certainly has some impact on the distribution of total optical depths, it should not pose a serious limitation while making reference calculations with DOM much cheaper[3].

The NWP SAF profiles do not contain any information on effective cloud particle sizes, which are required for RT calculations. Therefore, parameterizations have to be used. For effective radii of water cloud droplets, the parameterization of Martin et al. (1994) is used, which depends on the liquid water content and a droplet number concentration $N_C$. Here, we adopt either $N_C = 100 \, \text{cm}^{-3}$ or $N_C = 200 \, \text{cm}^{-3}$, which are typical values used in NWP models. For effective ice particle sizes we rely either on the parameterization by McFarquhar et al. (2003), which depends only on the ice content, or the one by Wyser (1998), which depends in addition on the temperature. All of these radius parameterizations can produce unrealistically small radii for low water/ice contents and under certain conditions also radii that are larger than the maximum radii RTTOV accepts.

---

[3]In RTTOV, DOM is called up to $n_z$ times instead of a single call if cloud fractions $0 < c < 1$ are encountered, where $n_z$ is the number of layers.

To reduce the impact of these cases, we limit the effective droplet radii to the range $[5\mu m, 25\mu m]$ and the effective ice particle radii to the range $[20\mu m, 60\mu m]$, as in Scheck et al. (2016). With these effective radii and the water / ice contents from the IFS data, extinction coefficient profiles for water and ice cloud layers ($\beta_w(z)$ and $\beta_i(z)$) can be computed using channel-specific conversion factors provided by RTTOV. The vertical integrals of $\beta_w(z)$ and $\beta_i(z)$ are the water and ice optical depths $\tau_w$ and $\tau_i$. From their distribution (Fig. 1) it is evident that there is a wide variety of water, ice and mixed-phase clouds with optical

depths up to several 100.

    In Tab. 1 the different versions of the profile data set used in this study are listed, which differ in the effective radius parameterizations, the cloud types present in the profiles and the vertical variation of the effective radii. The different parameterizations lead to significantly different mean effective radius distributions, as shown in Fig. 2. A smaller droplet concentration (Fig. 2b) leads to larger droplet radii than for the standard value of $N_C = 200\,\text{cm}^{-3}$ (Fig. 2a). The ice particle radii computed with

Wyser (1998) (Fig. 2d) show more spread and depend differently on the optical depth than those computed using McFarquhar et al. (2003) (Fig. 2c). The data sets with only water or only ice clouds allow for investigating these cloud types separately. The data sets in which the effective radius profile is replaced by its mean value in each profile is used to switch off the impact of vertical radius gradients.

    To compute top-of-atmosphere reflectances for the profiles in these data sets using DOM, not only the cloud variables but

also the sun and satellite angles and the surface albedo are required. For all the profiles, longitude, latitude and time are known, so these additional parameters could be determined. However, we follow a different approach, which increases the number of test cases significantly. For each profile, reflectances are computed for 64 angle combinations that are chosen randomly with the constraints $\alpha < 130°$, $\theta < 80°$ and $\theta_0 < 80°$, where $\alpha$ is the scattering angle with $\alpha = 0°$ meaning backscattering and $\alpha = 180°$ forward scattering. Moreover, reflectances are not computed for the surface albedo at the location of the profile,

but for each viewing geometry three reflectances are computed for albedo values $0$, $\frac{1}{2}$ and $1$. The three reflectances allow for calculating reflectance for arbitrary albedo values, as discussed in Sect. 3.3 of Scheck (2021). In total, $5000 \times 64 \times 3 = 960000$ reflectances are thus computed for each data set and RT method.

    Of course the effective radii obtained for the IFS profiles from parameterizations may not be fully realistic. Both features in the vertical profiles and the distribution of mean radii could differ from reality. For this reason, we will in addition consider

data from a regional model with a two-moment microphysics scheme that generates prognostic information on effective radii, which should be closer to reality (see next Section). However, as at least for the next years most models will be restricted to one-moment microphysics schemes and this study is aimed at generating synthetic satellite images for these models, it seems appropriate to use effective radii from parameterizations.

### 2.3   ICON hindcasts

For an additional evaluation of the results we use atmospheric profiles from a 30-day hindcast performed with the convection-permitting, regional ICON-D2 (ICOsahedral Non-hydrostatic, development version based on version 2.6.1; Zängl et al. 2015) model. This NWP model and the resulting profiles are independent and different from the IFS model which is the basis for the NWP SAF profiles used in the development of the MFASIS near-infrared approach, especially with regard to the cloud

**Table 1.** Profile data sets used in this study, based on the NWP SAF profiles. In some of the data sets the water or ice cloud mixing ratios, $Q_\mathrm{w}$ and $Q_\mathrm{i}$, are set to zero, the effective radii are set to a constant value in each profile (the mean value in the profile) or the radii are computed for a different droplet number concentration (specified in brackets) in the droplet size parameterization by Martin et al. (1994) or using the Wyser (1998) instead of the McFarquhar et al. (2003) parameterization for ice particles.

| data set | parameterizations | modifications |
|----------|-------------------|---------------|
| std | Martin[200 cm$^{-3}$], McFarquhar | - |
| rmod | Martin[100 cm$^{-3}$], Wyser | - |
| w-only | Martin[200 cm$^{-3}$], McFarquhar | $Q_\mathrm{i} = 0$ |
| i-only | Martin[200 cm$^{-3}$], McFarquhar | $Q_\mathrm{w} = 0$ |
| w-rconst | Martin[200 cm$^{-3}$], McFarquhar | $Q_\mathrm{i} = 0$, $r_\mathrm{eff,w}(z) =$const |
| i-rconst | Martin[200 cm$^{-3}$], McFarquhar | $Q_\mathrm{w} = 0$, $r_\mathrm{eff,i}(z) =$const |

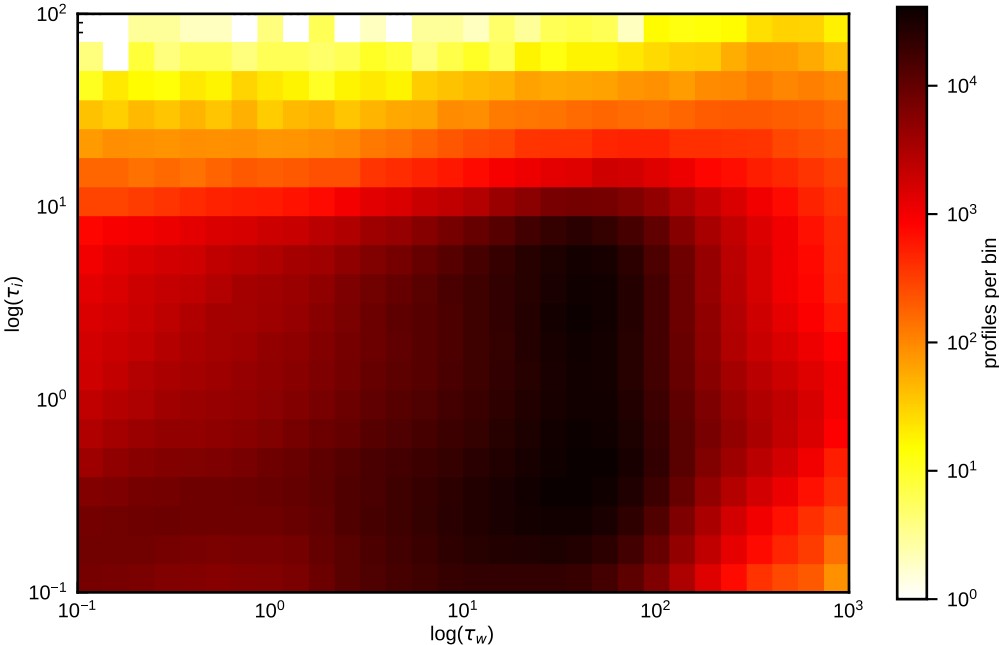

**Figure 3.** Distribution of water and ice cloud optical depths throughout the domain for all 30 days of the ICON-D2 hindcast period at 12 UTC (about $8.3 \times 10^6$ profiles).

microphysics scheme and the resulting effective cloud particle radii. The same ICON-D2 setup as in Geiss et al. (2021) is
used, with a domain covering Germany and parts of its neighbouring countries, a horizontal grid spacing of $2.1\,\mathrm{km}$ and 65 vertical levels. The 30-day hindcast is initialized once at 1 June 2020 00 UTC based on downscaled ICON-EU analysis initial conditions and uses hourly boundary conditions from ICON-EU analyses.

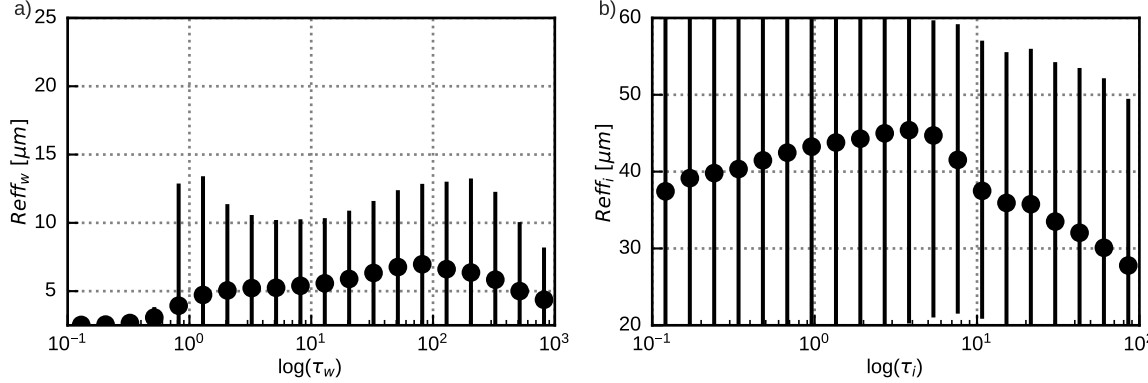

**Figure 4.** Mean vertically averaged effective particle radii (dots) in logarithmically spaced optical depth bins of the 30 days of the ICON-D2 hindcast period at 12UTC (about $8.3 \times 10^6$ profiles) for water clouds (a) and ice clouds (b). The vertical lines connect the 5th and the 95th percentiles for each bin.

The test period is characterized by various different summer time weather situations with a wide variety of clouds. As visible from Fig. 3), most of the clouds are limited to $\tau_w < 100$ and $\tau_i < 10$. The even thicker clouds present in the IFS profile data

sets are probably related to the tropics and not present in the ICON-D2 domain.

An important difference, compared to the IFS-based NWP SAF profiles, is that the ICON-D2 hindcast employs a two-moment microphysics scheme (Seifert and Beheng, 2006). This microphysics scheme directly provides effective radii that should in principle be more realistic than the radii computed with parameterizations. For this reason the lower limit for effective droplet radii applied before running RT calculations is reduced to $2.5\,\mu$m for the ICON hindcasts.

The effective radii from the two-moment scheme (Fig. 4) show qualitatively different dependencies on the optical depths, compared to those obtained with parameterizations (Fig. 2). For the two-moment calculations the mean effective droplet radius reaches a maximum at $\tau_w = 100$ and decreases for higher optical depths, whereas for the parameterization there is a further increase for $\tau_w > 100$. The mean effective ice particle radii from the two-moment scheme are more similar to the Wyser (Fig. 2c) than to the McFarquhar (Fig. 2d) results, with radii increasing at smaller decreasing at higher optical depths. However,

for the two-moment scheme the maximum mean effective radius is reached already at $\tau_i = 4$ (not $\tau_i = 30$ as for Wyser). Even more obvious differences between parameterized and two-moment radii are found for the spread. The parameterizations mostly depend on quantities strongly correlated with the optical depth, like LWC and IWC, and are therefore quite well-defined functions of the optical depth with a small spread. The only exception is the Wyser parameterization (Fig. 2d), which has an additional dependency on temperature and therefore a larger spread. In contrast, the two-moment radii always show a spread

that is considerably larger than for the parameterizations for both water and ice clouds, as is to be expected for more realistic radii.

## 3 Selecting input parameters

### 3.1 Extending the MFASIS approach

For visible channels it is sufficient to characterize the idealised profiles by only four numbers, optical depths and effective
particle radii for a water and an ice cloud at fixed heights. As discussed in Sect. 2.1, these idealised profiles are too simplistic
for near-infrared channels. Therefore, additional parameters have to be included to account for the clear sky absorption, the
impact of vertically varying effective radii and the fact that the uppermost part of mixed-phase clouds is often dominated by
liquid water.

As a minimal extension to account for vertically varying effective radii, the one-layer ice and water clouds are replaced
by two-layer clouds with different effective radii in the upper and the lower layer. Moreover, for a better representation of
mixed-phase clouds we allow for the presence of ice in the two-layer water cloud, which can therefore either be a water or a
mixed-phase cloud. In contrast, the ice cloud located above this mixed-phase cloud is assumed to be free of liquid water. Thus,
in total, six optical depths $\tau_C^L$ and six effective particle radii $r_C^L$ have to be specified to fully define the clouds in the idealised
profile. Here, $L \in [\text{u;l}]$ for the upper/lower layers, $C = \text{i}$ for the pure ice cloud, $C = \text{w}$ for the water content and $C = \text{wi}$ for
the ice content of the mixed-phase cloud.

When the geometric height at the top, $z_C^{L,\text{top}}$, and the bottom, $z_C^{L,\text{bot}}$ of each layer is known, the optical depths can be
computed as

$$
\tau_C^L = \int_{z_C^{L,\text{bot}}}^{z_C^{L,\text{top}}} \beta_C(z)\,\mathrm{d}z
\tag{2}
$$

from the profile of the extinction coefficient $\beta_C(z)$ computed from the NWP model output using the satellite channel-dependent
factors provided by RTTOV (as part of the regression coefficient files). The mean effective particle radii are computed as

$$
r_C^L = (\tau_C^L)^{-1} \int_{z_C^{L,\text{bot}}}^{z_C^{L,\text{top}}} r_{\text{eff},C}(z)\,\beta_C(z)\,\mathrm{d}z
\tag{3}
$$

from the effective radius profile $r_{\text{eff},C}(z)$ that is either included in the NWP model output or computed using one of the
parameterizations listed in Sect. 2.2. For computing the mixed-phase cloud top height in the NWP profile, we use the cumulative
optical depth

$$
\tau_C^{\text{cml}}(z) = \int_z^{z_{\text{toa}}} \beta_C(z)\,\mathrm{d}z,
\tag{4}
$$

where $z_{\text{toa}}$ is the height of the uppermost level of the model profile. The mixed-phase cloud top height, $z_{\text{wi}}^{\text{u,top}}$, is then defined
to be the height at which $\tau_{\text{w}}^{\text{cml}}(z)$ exceeds a threshold value of 1. To avoid gaps between the integration ranges $z_C^{l,\text{top}} = z_C^{u,\text{bot}}$
and to cover the full profile, we adopt $z_{\text{w}}^{u,\text{top}} = z_{\text{i}}^{u,\text{top}} = z_{\text{toa}}$ and $z_{\text{w}}^{l,\text{bot}} = z_{\text{wi}}^{l,\text{bot}} = z_{\text{sfc}}$, where $z_{\text{sfc}}$ is the height of the lowest

level of the model profile[4]. The only geometric heights not defined so far are those that determine how the two-layer clouds are split in an upper and a lower part, $z_i^{u,bot}$ and $z_w^{u,bot}$. A parameterization for the cloud splitting that computes the optical depths of the two parts, $\tau_C^l$ and $\tau_C^u$, from the total optical depth of the cloud, $\tau_C = \tau_C^l + \tau_C^u$, will be discussed in Sect. 3.3. Based on these optical depths, $z_i^{u,bot}$ and $z_w^{u,bot}$ can be determined.

The impact of clear sky absorption by the well-mixed trace gases carbon dioxide ($CO_2$) and methane ($CH_4$) depends on the air mass above the cloud and, in case of semi-transparent clouds, also on the air mass above the ground. The latter can be quantified by the cloud top pressure, $p_{ct}$, and the surface pressure, $p_{sfc}$. Of course, $p_{ct}$ and $p_{sfc}$ cannot exactly quantify the clear sky impact for complex multi-layer cloud situations, but should still be useful for an approximate description. In this study, the cloud top pressure is defined as the pressure at which the total (water plus ice cloud) optical depth exceeds a threshold value of $\min(\frac{1}{2}\tau_t, 1)$, where $\tau_t$ is the column-integrated total optical depth of all water and ice cloud layers. This definition presents a good compromise to prevent on the one hand that very thin, high clouds above thicker clouds trigger the cloud top detection and to avoid on the other hand that the cloud top is detected too deep inside the cloud. Instead of $p_{ct}$, we will use the dimensionless ratio $f_{ct} = p_{ct}/p_{sfc}$ as input parameter for neural networks. Also, absorption by water vapour has some influence on $1.6\mu m$ SEVIRI reflectances, in particular for high albedo values and large zenith angles. Water vapour is not well-mixed and therefore its impact can not just be quantified by $p_{ct}$ and $p_{sfc}$. As the impact is relatively weak, it is sufficient to use just one parameter to describe the water vapour profile, the vertically integrated water vapour content,

$$\text{IWV} = \int_{z_{sfc}}^{z_{toa}} \rho(z)q_v(z)\mathrm{d}z, \tag{5}$$

where $\rho$ is the density and $q_v$ is the specific humidity. Instead of IWV, we will use the normalized

$$\text{nIWV} = \frac{(\text{IWV} - \text{IWV}_{min}(p_{sfc}))}{(\text{IWV}_{max}(p_{sfc}) - \text{IWV}_{min}(p_{sfc}))} \tag{6}$$

as input parameter for neural networks. This ensures that IWV remains within the range accepted by RTTOV. Details on the fit functions $\text{IWV}_{min}$ and $\text{IWV}_{max}$ can be found in Appendix B.

The definitions provided so far allow for computing the neural network input parameters $p_{ct}$, $p_{sfc}$, nIWV, $\tau_C^L$ and $r_C^L$ from NWP profiles, a step required before reflectances can be computed with the network. For generating the synthetic training data for the network, the inverse step is required: For given network input parameters we need to define full idealised vertical profiles, as the latter are required by DOM to compute the corresponding reflectances. These idealised profiles have to contain the four cloud layers with the desired $\tau_C^L$, $r_C^L$, have the correct IWV, the correct cloud top pressure $p_{ct}$ (according to the definition given above) and have to start at the correct surface pressure $p_{sfc}$. The geometric thickness of the cloud layers should not be very important for the reflectance. For the sake of simplicity, we modify an IFS standard atmosphere such that one of the pressure levels matches the cloud top pressure and that the surface pressure has the desired value. Then only the four layers around $p_{ct}$ are filled with $\tau_C^L$, $r_C^L$, as shown for an example in Fig. 5, which illustrates also the integration ranges used to

---

[4]A better definition for the cloud bottom level could have been used here, but as integrating over cloud free layers near the surface will would not change Eqns. 2 and 3, using $z_{sfc}$ is sufficient.

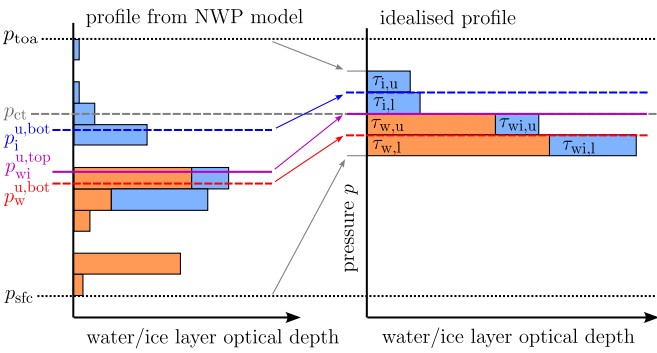

**Figure 5.** Schematic example of a complex profile from a NWP model (left) and the corresponding idealised profile (right), in which only four layers are filled with clouds. The dashed blue and red lines separate the upper and lower parts of the ice and the mixed phase cloud, respectively. The purple line separates pure ice from mixed phase ice. The dashed gray line indicates the cloud top pressure and the dotted black line the surface pressure. The pressure labels correspond to the geometric heights $z$ with the same indices defined in the text. Please note: The thickness of the layers is exaggerated, the four cloud layers in the idealised profile are all close to $p_{\mathrm{ct}}$.

compute the $\tau_C^L$ and $r_C^L$ from a NWP model profile. A standard water vapour profile is scaled such that the correct IWV results.
Details on how idealised profiles are computed from the input parameters can be found in Appendix A.

The computation of reflectances for the idealized profiles with two-layer clouds including a mixed phase, surface and cloud top pressures (as shown in the right half of Fig. 5) and the scaled water vapour background, now requires in total 16 parameters. This assumes that the transmittances of the upper and lower water and ice cloud layers can be computed as a function of their overall optical depths $\tau_{\mathrm{w}}$ and $\tau_{\mathrm{i}}$ (see parameterization described in Sect. 3.3). For these input parameters we will use the
abbreviation

$$\mathbf{p} = (\theta, \theta_0, \Delta\phi,\ p_{\mathrm{sfc}}, f_{\mathrm{ct}}, \mathrm{nIWV},\ \tau_{\mathrm{w}}, r_{\mathrm{w}}^{\mathrm{u}}, r_{\mathrm{w}}^{\mathrm{l}},\ \tau_{\mathrm{i}}, r_{\mathrm{i}}^{\mathrm{u}}, r_{\mathrm{i}}^{\mathrm{l}},\ \tau_{\mathrm{wi}}^{\mathrm{u}}, \tau_{\mathrm{wi}}^{\mathrm{l}}, r_{\mathrm{wi}}^{\mathrm{u}}, r_{\mathrm{wi}}^{\mathrm{l}}). \tag{7}$$

Although also required as an input parameter for DOM, the albedo $A$ was not included in $\mathbf{p}$ because it is not an input variable for the neural networks considered in this study (see explanation in Sec.4.1). In the rest of Sect. 3 only results for $A = 0.5$ will be discussed. Results for different albedo values will be presented in Sect. 5. As will be discussed in Sect. 3.4, the splitting of
the water cloud is also used for the ice content of the mixed-phase cloud. Accordingly, not just one $\tau_{\mathrm{wi}}$ but the optical depths for the upper and the lower part have to be specified.

Using the idealised profiles instead of the NWP model profiles as input for DOM will lead to reflectances errors, which should decrease with the number of parameters used to characterize the idealised profiles. In the following, we will discuss the relative importance of the new input parameters by using different profile data sets (see Tab. 1) and different profile
simplifications. Which profiles were used as input for DOM to compute reflectances is indicated by the following indices:

| DOM | Full non-idealised profiles (reference solution) |
| --- | --- |
| 2Lay,mp | Idealised profiles characterized by the 16 input parameters in $\mathbf{p}$ |
| 2Lay | like 2Lay,mp, but all ice is moved to the pure ice cloud (12 parameters) |
| 1Lay | Like 2Lay, but using only one layer per cloud (10 parameters) |
| 1Lay,fix | Like 1Lay, but cloud top and base set to fixed heights, no IWV (7 parameters) |

For 2Lay, 1Lay and 1Lay,fix the integration ranges are chosen such that all ice ends up in the pure ice cloud ($z_{\mathrm{i}}^{\mathrm{l,bot}} = z_{\mathrm{wi}}^{\mathrm{u,top}} = z_{\mathrm{sfc}}$). The 1Lay,fix setup corresponds to the original MFASIS from Scheck et al. (2016) and also the same fixed cloud top and base heights are adopted.

## 3.2 Impact of surface pressure and cloud top height

To quantify the impact of taking $p_{\mathrm{ct}}$, $p_{\mathrm{sfc}}$ and IWV into account, it is helpful to exclude other sources of reflectance error. For this purpose, the data sets 'w-rconst' and 'i-rconst' (see Tab. 1) are used, which contain only clouds of one phase with vertically constant effective radii. Errors related to mixed-phase clouds and vertical radius gradients are thus excluded, but errors related to the simplification of the vertical cloud / clear sky structure are present. These errors should be smaller when in the idealised profiles $p_{\mathrm{ct}}$ and $p_{\mathrm{sfc}}$ have the same value as in the original profile, so that approximately the same air masses above and below the cloud top influence the reflectance by molecular absorption and Rayleigh scattering. In the absence of vertical radius gradients it is sufficient to consider one-layer clouds and we therefore compare reflectances computed with the cloud top and surface pressures from the original profile, $R_{\mathrm{1Lay}}$, to the ones computed with fixed cloud top and surface pressures, $R_{\mathrm{1Lay,fix}}$, which uses the same idealised profiles as in Scheck et al. (2016). The mean absolute errors $|R_{\mathrm{1Lay}} - R_{\mathrm{DOM}}|$ and $|R_{\mathrm{1Lay,fix}} - R_{\mathrm{DOM}}|$ with respect to the reference solution for both water and ice clouds are shown in Fig. 6 as a function of optical depth and the maximum of the zenith angles.

The errors for the water clouds (Fig. 6a, b) are higher than for the ice clouds (Fig. 6c, d), as for clouds located at lower heights the air masses and their impact on reflectance are higher. Also larger zenith angles lead to a stronger influence of molecular absorption and Rayleigh scattering and thus higher errors in Fig. 6, as they increase the photon path lengths. For both data sets a clear error reduction (Fig. 6a to b, c to d) is visible when $p_{\mathrm{ct}}$, $p_{\mathrm{sfc}}$ and $IWV$ are taken into account. In particular, errors at higher optical depths are reduced ($\tau_{\mathrm{w}} > 100$, $\tau_{\mathrm{i}} > 10$), but also at intermediate optical depths ($\tau_{\mathrm{w}} = 10\ldots100$, $\tau_{\mathrm{i}} = 1\ldots10$) significant error reduction can be observed. For ice clouds, the remaining mean absolute errors are of order $\mathcal{O}(10^{-3})$, i.e. very small (Fig. 6d). For water clouds, the errors at $\tau_{\mathrm{w}} > 100$ are reduced to similarly low levels (Fig. 6b). For water clouds of intermediate optical depths, errors are reduced considerably, but mean absolute errors around 0.01 can still be found when $p_{\mathrm{ct}}$ and $p_{\mathrm{sfc}}$ are taken into account. However, these errors are already in an acceptable range. In fact, for both data sets, only $15\,\%$ of the original mean absolute error (MAE) and $25\,\%$ of the extreme errors (P99) remain (see Tab. 3).

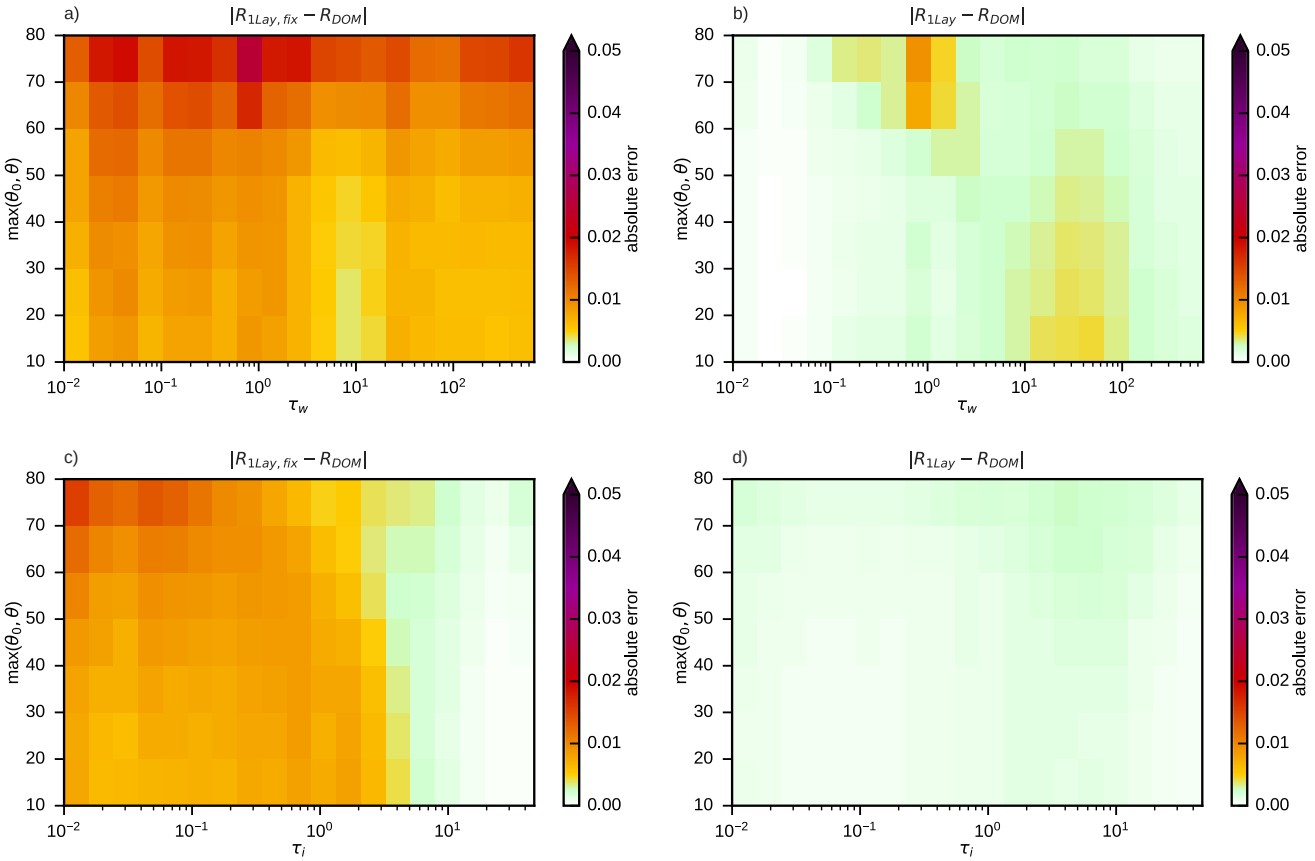

**Figure 6.** Mean absolute reflectance errors for simplified profiles without (a,c) and with (b,d) taking cloud top and surface pressure into account with respect to the DOM reference solution for the full profiles of the 'w-rconst' (a,b) and 'i-rconst' (c,d) data sets in bins defined by optical depth and the maximum of solar and satellite zenith angles.

### 3.3 Optimizing two-layer clouds

In Sect. 3.1 two-layer clouds were introduced as a simple way to approximate the effect of vertical effective radius gradients. It still has to be defined how the clouds in the idealised profiles should be split up in an upper and a lower layer. For optically thin clouds the probability of a photon to be scattered in the upper or the lower half (in terms of optical depth) of the cloud should be similar, so it makes sense to split them such that $\tau_C^u = \tau_C^l$ (where $C = $ w for water and $C = $ i for ice clouds) and compute mean effective radii for the upper and lower half. For denser clouds the contribution of layers deeper in the cloud to the top-of-atmosphere reflectance should be smaller due to absorption. To resolve the effective radius gradient in regions where it has the strongest impact on the reflectance, it should in this case be more appropriate to split the clouds in a thinner upper part and a thicker lower part. The optimal optical depth to split at depends of course also on the vertical effective radius profile. We assume that a parameterization for a splitting factor $f_C = \frac{\tau_C^u}{\tau_C}$ can be found that works reasonably well for many different

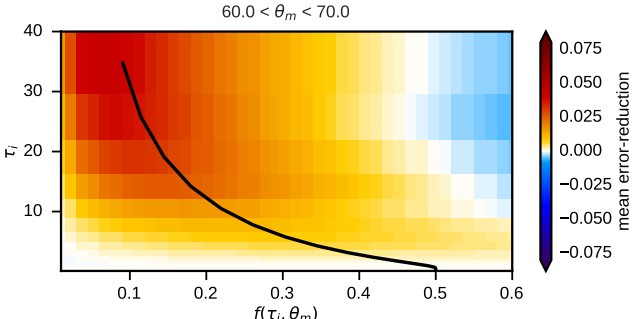

**Figure 7.** Mean absolute reflectance error reduction $|R_{1\mathrm{Lay}} - R_{\mathrm{DOM}}| - |R_{2\mathrm{Lay}} - R_{\mathrm{DOM}}|$ achieved by using two-layer instead of one-layer clouds for the 'i-only' data set for different values of the layer splitting factor $f_i$ and the optical depth $\tau_i$ in the zenith angle bin $60° < \theta_m < 70°$. Positive values mean the two-layer approach is better than the one-layer approach. The splitting factor parameterization as defined in Eq. 8 is visualized by the white line.

effective radius profiles, where $\tau_C = \tau_C^{\mathrm{u}} + \tau_C^{\mathrm{l}}$ is the total optical depth of the two-layer cloud. From the argumentation above, the parameterization should produce a value of $\frac{1}{2}$ for small optical depths and decline with increasing $\tau_C$. However, also the zenith angles should play a role, because they influence the path length in the cloud and thus also the probability of a photon
to be absorbed. For the sake of simplicity, we use the single parameter $\theta_m = \max(\theta_0, \theta)$, the maximum of the two zenith angles, to quantify the zenith angle dependence. For determining a reasonable function $f_C(\tau_C, \theta_m)$, the mean error reduction $|R_{1\mathrm{Lay}} - R_{\mathrm{DOM}}| - |R_{2\mathrm{Lay}} - R_{\mathrm{DOM}}|$ was computed for the 'w-only' and 'i-only' data sets and many different bins defined by $f_C$, $\tau_C$ and $\theta_m$. As an example, the case $60° < \theta_m < 70°$ for ice clouds is shown in Fig. 7. It is obvious that for low optical depths it does not make much of a difference where exactly the cloud is split and that with increasing optical depth the error
can be reduced more strongly when smaller values of $f_C$ are chosen.

A function producing near-optimal values for the splitting factor is given by

$$f_C(\tau_C, \theta_m) = \frac{1}{2} - \left[ \frac{1}{2} - f_{hi}(\theta_m) \right] \times \frac{1}{2} \left[ 1 - \cos(\pi t_C(\tau_C)) \right], \tag{8}$$

where

$$t_C(\tau_C) = \min\left(1, \max\left(0, \frac{\log(\tau_C) - \log(\tau_{C,lo})}{\log(\tau_{C,hi}) - \log(\tau_{C,lo})}\right)\right) \tag{9}$$

with $\tau_{i,lo} = 0.6$, $\tau_{i,hi} = 75$ for ice clouds and $\tau_{w,lo} = 1.0$, $\tau_{w,hi} = 500$ for water clouds, and the zenith angle dependent term

$$f_{hi}(\theta_m) = f_{zen} - \frac{\theta_m}{90°} \Delta f_{hor} \tag{10}$$

with $f_{zen} = 0.1$ and $\Delta f_{hor} = 0.05$. The constants used in the definition of $f_C(\tau_C, \theta_m)$ are chosen such that in plots like Fig. 7 for all angle bins and both water and ice clouds the parameterization yields a value of the splitting factor that is close to maximising the error reduction, as illustrated by the example in Fig. 7 (black line). For both water and ice clouds the
parameterized splitting factors start at $\frac{1}{2}$ for low optical depths, decrease with increasing depth and saturate at the same value

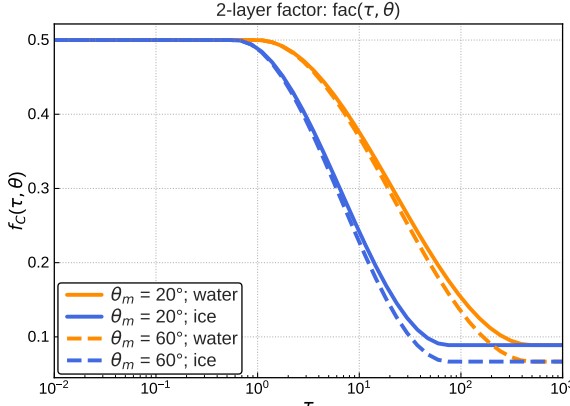

**Figure 8.** The two-layer splitting factor parameterization $f_C(\tau_C, \theta_m)$ for water (orange) and ice clouds (blue) as a function of the optical depth and for two different values of the zenith angle parameter, $\theta_m = 20°$ (solid) and $\theta_m = 60°$ (dashed).

for high optical depths (Fig. 8). For ice clouds, which are optically considerably thinner than the water clouds (see Fig. 1), the decline takes place at lower optical depths.

For testing the two-layer approach, the data sets 'w-only' and 'i-only' are considered, which do not contain mixed-phase clouds but have vertically varying effective radii. For both data sets the mean absolute errors for the one-layer approach with surface and cloud top pressure taken into account are considerably worse than for the corresponding data sets without vertical radius gradients (compare Fig. 6b to Fig. 9a, Fig. 6d to Fig. 9c, and see Tab. 3). In particular for optical depths larger than 10 reflectance errors exceeding 0.05 can be found. Switching from the one-layer to the two-layer approach with the parameterized splitting factors reduces the error considerably (compare Fig. 9a to b, c to d) at these moderate to high optical depths. The remaining errors are around 0.01 or lower in most parts of the parameter space, only for large zenith angles and very high optical depths somewhat larger values con be found (Fig. 9b,d).

## 3.4 Accounting for mixed-phase clouds

So far, only data sets without mixed-phase clouds have been used. For the 'std' data set including mixed-phase clouds the two-layer approach without special treatment of mixed-phase clouds results in strongly increased errors. The mean absolute reflectance error, now computed for bins of the total optical depth of all water and ice layers in the column, $\tau_t$, and the zenith angle parameter $\theta_m$ can exceed 0.05 for high optical depths (Fig. 10a), which is considerably higher than errors for pure water (Fig. 9b) and pure ice clouds (Fig. 9d). The misrepresentation of mixed-phase clouds thus can cause errors that are similar in size to the ones related to not taking vertical effective radius gradients into account. By allowing for mixed-phase ice in the two layers of the mixed-phase cloud in the idealised profiles, as discussed in Sect. 3.1, these errors can be reduced significantly. As shown in Fig. 10b, the mean absolute reflectance errors are around 0.01 or lower in most of the parameter space.

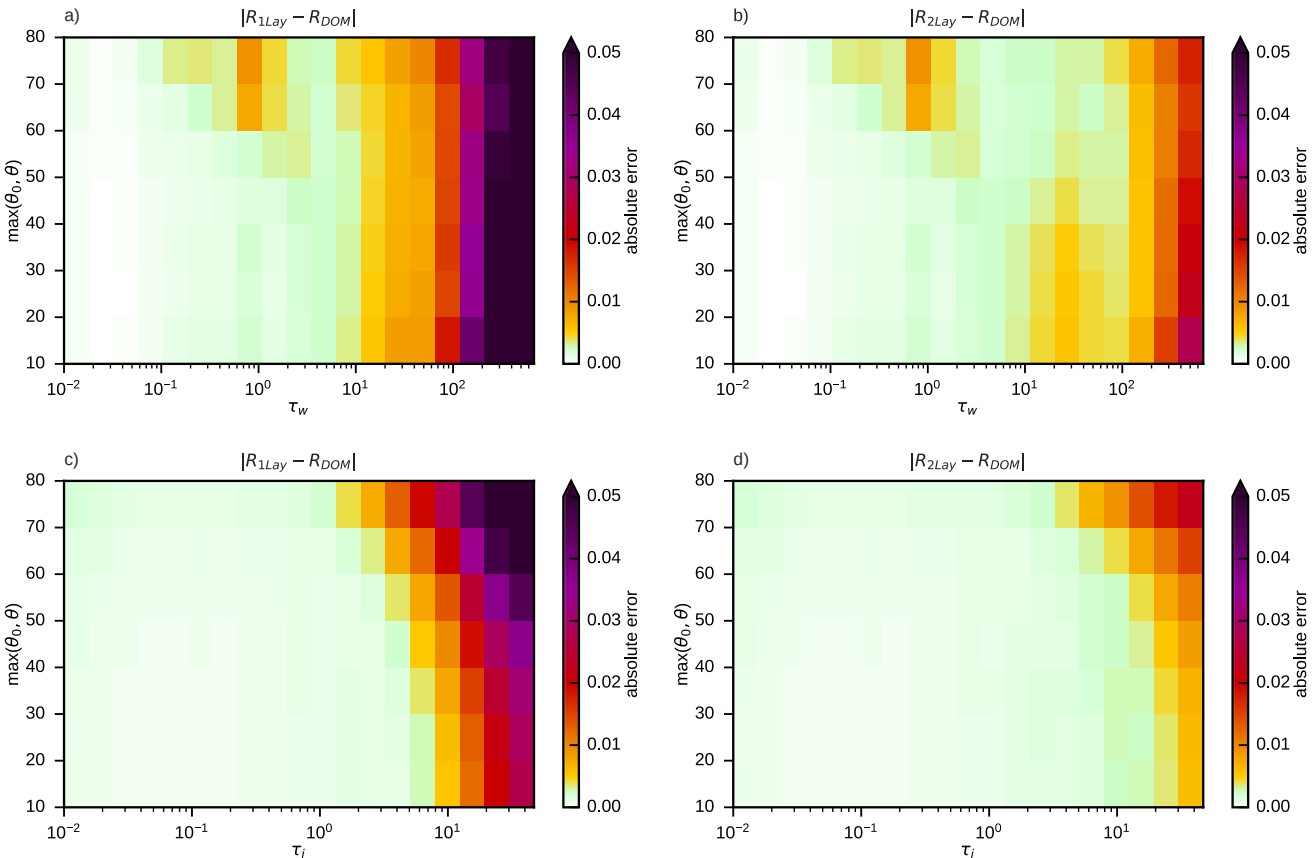

**Figure 9.** Mean absolute reflectance errors for simplified profiles with one-layer (a,c) and two-layer clouds (b,d) with respect to the DOM reference solution for the full profiles of the 'w-only' (a,b) and 'i-only' (c,d) data sets in bins defined by optical depth and the maximum of solar and satellite zenith angles. In all cases cloud top and surface pressure were taken into account.

## 3.5 A simple bias correction

Most of the remaining mean absolute error in Fig. 10b is actually not a random error, but related to a mean error or bias, as evident from Fig. 11a. A part of the bias may depend on details of the profile data set and the effective radius parameterizations used here. However, most of it may be independent of the details and directly related to the profile simplifications and therefore would still be present for a 'perfectly realistic' data set. To confirm this conjecture it seems worthwhile to develop a simple bias correction to remove most of the remaining mean errors for the 'std' data set, and investigate later on if the correction is helpful for other data sets. Due to the clear structure of the bias it is easy to derive a simple function of the total optical depth and the zenith angle parameter $\theta_m$, which reproduces the mean error. Here we use a Gaussian shaped correction

$$\Delta R_{\mathrm{bc}}(\tau_{\mathrm{t}}, \theta_{\mathrm{m}}) = \frac{m + n \times (\theta_{\mathrm{m}}/90°)^k}{\sqrt{2\pi}\ \tau_{\mathrm{t}}\sigma\ \exp\left(-\frac{(log(\tau_{\mathrm{t}})-\mu)^2}{2\sigma^2}\right)} \tag{11}$$

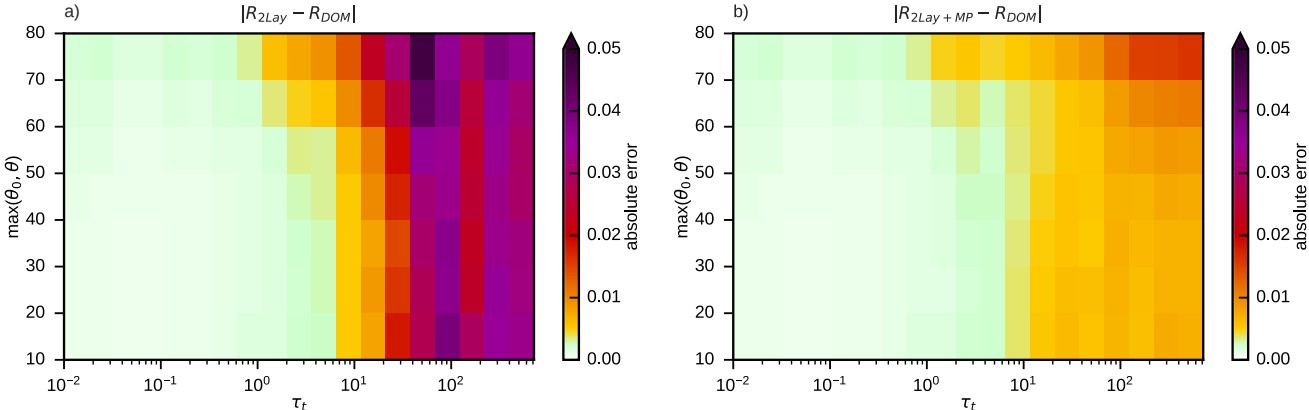

**Figure 10.** Mean absolute reflectance errors for simplified profiles without (a) and with (b) special treatment of mixed phase clouds, with respect to the DOM reference solution for the full profiles of the 'std' data set in bins defined by total optical depth and the maximum of solar and satellite zenith angles. In all cases cloud top and surface pressure were taken into account and two-layer clouds were used.

with the parameters ($\sigma = 1.1, \mu = 7, m = 12, n = 30, k = 3.5$) chosen such, that the correction predominantly acts for $\tau_t$ larger than approximately 50. As Figure 11b shows, applying the bias correction to the 'std' data set removed most of the bias and reduces the MAE by 25 % and the P99 by about 20 % (see Tab. 3). Also for the 'rmod' data set using different effective radius parameterizations, the bias correction (Fig. 11c) still reduces the MAE and the P99 by about 20 % (see Tab. 3). The fact that the bias correction reduces errors for both data sets indicates that it does not depend strongly on details of the 'std' data set and that it seems to correct a more generic error related to the profile simplification.

## 4  Network training

As discussed in the previous section, replacing vertical profiles from NWP model runs by the corresponding idealised profiles characterized by the same parameters results in only small reflectance errors. These profile parameters are thus in principle suitable as input parameters for a neural network to predict reflectances.

### 4.1  Setup

The function to be learned by the neural network is similar but due to the additional parameters somewhat more complex than in case of the visible channel investigated in Scheck (2021). Therefore, we will investigate for the near-infrared channel if networks of a similar or slightly larger size as considered for the visible channel can be trained with similar training settings to achieve comparable reflectance errors. In this section only errors due to imperfect training of the networks will be considered, but not the errors caused by simplifying the vertical structure that have been discussed in the previous section. Therefore, the

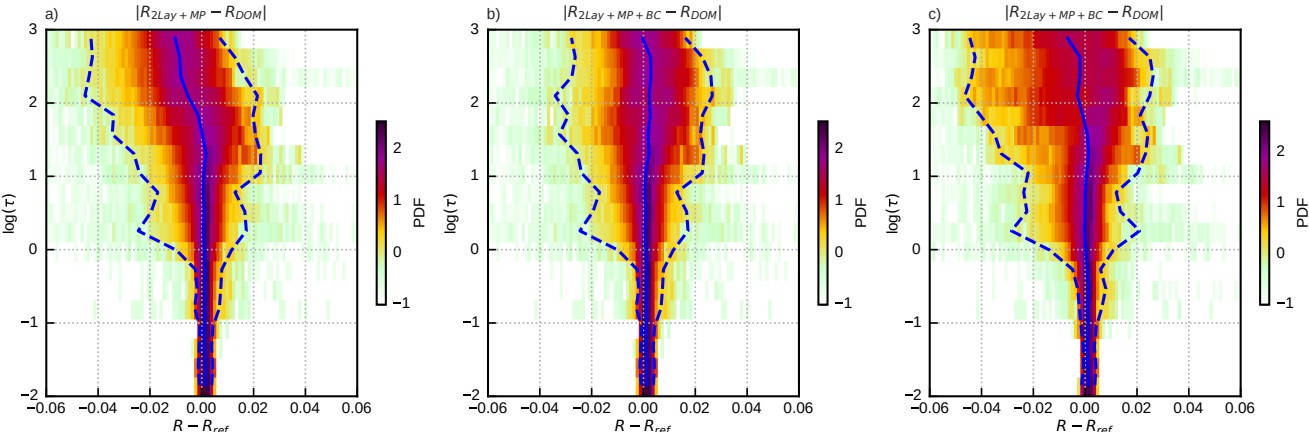

**Figure 11.** Error PDFs (shaded) showing the deviation of different reflectance estimations ($R$) from DOM reference computations ($R_{\text{ref}}$) as a function of optical depth ($log(\tau)$). These are not 2D-PDFs – each horizontal cut through these images represents the 1D-PDF for the optical depth bin indicated on the vertical axis. The blue, dashed lines depict the 1st (left) and 99th (right) percentile of the error, while the blue, solid line shows the mean error. The error histograms are shown for 2Lay+MP (a) of the standard data set, as well as 2Lay+MP+BC for the standard (b) and rmod (c) data sets.

$R_{2\text{Lay,mp}}$ reflectances computed with DOM for idealised profiles with randomly chosen parameters will serve as training and validation data. About 30 million samples, i.e. tuples of input parameters and reflectances, were generated for this purpose.

In Fig. 12 an example for the network structure is shown. The nodes of the input layer correspond to the elements of **p** (defined in Eq. 7) describing the idealised profiles and the sun/satellite geometry. The range applied for the parameters is listed in Tab. 2. There is no input node for the surface albedo, as the latter is treated in a different way. As discussed in more detail

in Scheck (2021), in plane-parallel RT the reflectance for an arbitrary albedo value can be computed exactly, if reflectances for three different values are known. If the network is trained to generate reflectances for three different albedo values, albedo is thus not required as input parameter and errors resulting from an imperfect representation of the albedo dependence by the network can be avoided. Following Scheck (2021), the reflectance for albedo zero, $R(0)$, and the reflectances differences $D_{\frac{1}{2}} = R(\frac{1}{2}) - R(0)$ and $D_1 = R(1) - R(\frac{1}{2})$ are chosen as output parameters, where $R(A)$ is reflectance as a function of

surface albedo $A$. These differences are used instead of $R(\frac{1}{2})$ and $R(1)$ because the computation of reflectance for an arbitrary $A$ requires $R(1) > R(\frac{1}{2}) > R(0)$ to avoid numerical problems and by using softplus as activation function for the output nodes the constraints $R(0) > 0$, $D_{\frac{1}{2}} > 0$ and $D_1 > 0$ are automatically fulfilled. Following the Tensorflow standard approach, the root mean squared error (RMSE) of the all network output variables, $\epsilon_{\text{TF}}$, is minimized in the training.

In contrast to Scheck (2021), we will not explore the full parameter space of training and network related settings, but show

only that for a plausible choice of training and network structure settings sufficiently accurate networks result. Only the most relevant setting for the network structure is varied, which is the total number of parameters. For the visible channel, networks with 2000 parameters resulted already in acceptable reflectance errors that were somewhat smaller than the interpolation

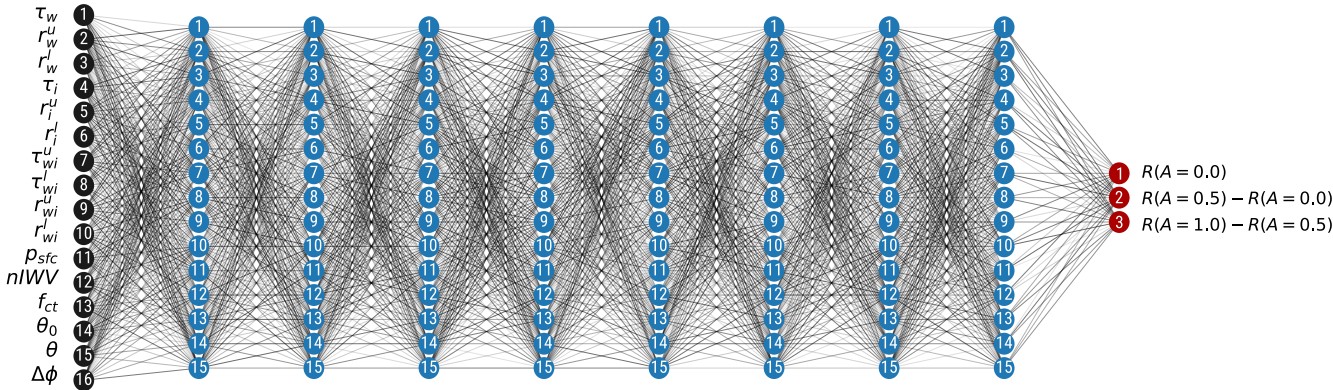

**Figure 12.** Structure of the smallest network considered, with 16 input nodes (black), 8 hidden layers with 15 nodes per layer and in total 1983 weight and bias parameters. As activation functions (see Sect. 2.1.3) CSU (Eq. 1) is used for the hidden layers (blue nodes) and softplus for the output layer (red nodes). The lines in different shades of gray symbolise the network weights.

**Table 2.** List of input parameters for the neural networks, their abbreviation and ranges.

| Parameter | Abbreviation | | Range |
|---|---|---|---|
| water optical depth | $\tau_{\mathrm{w}}$ | $\in$ | $[10^{-3}, 10^{3}]$ |
| ice optical depth | $\tau_{\mathrm{i}}, \tau_{\mathrm{wi}}^{\mathrm{u}}, \tau_{\mathrm{wi}}^{\mathrm{l}}$ | $\in$ | $[10^{-3}, 10^{2}]$ |
| water effective radius | $r_{\mathrm{w}}^{\mathrm{u}}, r_{\mathrm{w}}^{\mathrm{l}}$ | $\in$ | $[1\,\mu\mathrm{m}, 26\,\mu\mathrm{m}]$ |
| ice effective radius | $r_{\mathrm{i}}^{\mathrm{u}}, r_{\mathrm{i}}^{\mathrm{l}}, r_{\mathrm{wi}}^{\mathrm{u}}, r_{\mathrm{wi}}^{\mathrm{l}}$ | $\in$ | $[5\,\mu\mathrm{m}, 60\,\mu\mathrm{m}]$ |
| surface pressure | $p_{\mathrm{sfc}}$ | $\in$ | $[600\,\mathrm{hPa}, 1050\,\mathrm{hPa}]$ |
| dimensionless cloud-top pressure | $f_{\mathrm{ct}}$ | $\in$ | $[0, 1]$ |
| normalized column-integrated water vapour | nIWV | $\in$ | $[0, 1]$ |
| zenith angle | $\theta, \theta_0$ | $\in$ | $[0°, 85°]$ |
| scattering angle | $\Delta\phi$ | $\in$ | $[0°, 180°]$ |

errors in the LUT version, and for 5000 parameters the errors were negligible, compared to the error caused by the profile simplification (Scheck, 2021). Here we investigate networks with 1983 (NN2k), 5053 (NN5k) and 8035 (NN8k) parameters, which are trained for 4000 epochs using $14 \times 10^{6}$ training samples.

## 4.2 Results

From Fig. 13, which shows the evolution of the RMSE of the output parameters, $\epsilon_{\mathrm{TF}}$, during the training of the three networks, it is obvious that for all of them errors of several $10^{-3}$ are reached for both the training and the validation data sets. Due to the early stopping strategy the errors for the validation data set are somewhat smaller than for the training data set. The final errors

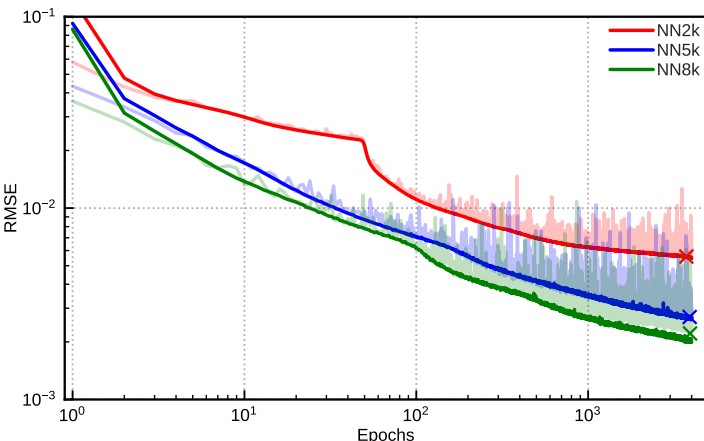

**Figure 13.** Evolution of the training (solid color) and validation (light color) RMSE during the course of the training for a networks with 1983 (NN2k, red), 5053 (NN5k, blue) and 8053 (NN8k, green) parameters. The final validation RMSE values for each network (indicated by the crosses) are $5.6 \cdot 10^{-3}$ (NN2k), $2.7 \cdot 10^{-3}$ (NN5k) and $2.2 \cdot 10^{-3}$ (NN8k).

provided in Fig. 13 suggest that a number of parameters between 2000 and 5000 may be a reasonable choice and not much accuracy is gained by increasing the number of parameters further. It should be noted that the NN8k network shows first signs of overfitting – a small gap appears between the training data and the validation data after about 2000 epochs. Further training of NN8k would thus require larger amounts of training data. For the smaller networks the amount of training data seems to be sufficient. The choice of the network size should of course depend also on the computational costs for evaluating networks of

different sizes. Benchmarks for different network sizes and activation function have already been investigated for the optimized FORNADO inference code (see Fig. A12 in Scheck 2021). Evaluating the 1983, 5053 and 8035 parameter networks considered here takes $0.14\,\mu$s, $0.30\,\mu$s and $0.51\,\mu$s per column, respectively, on a core of a standard Intel Xeon 8358 server CPU running at $3.3$GHz. For comparison, the look-up table based MFASIS (using the LUT with $N_k = N_l = 3$, see Tab. 3 of Scheck et al. 2016) takes $1.58\,\mu$s and the DOM implementation in RTTOV (with 16 streams) takes about $17$ms per column on the same

CPU. These timings are only for the computation of reflectances from the input parameters[5] listed in Tab. 2. Although some additional effort is required for computing the network input parameters from the NWP profiles, it should be possible to process e.g. $10^6$ model columns in several CPU-seconds with these networks.

     The networks and training procedures discussed here are not yet fully optimized. Further gains in accuracy could be expected e.g. from tuning the learning rate, the batch size and the number of hidden layers. Also regularization techniques could make

the training more efficient and using a lower precision data type, e.g. 16 bit instead of 32 bit floating point values, could speed up training and inference. However, such tuning efforts are not in the focus of this study and already in their current state the neural networks seem to be sufficiently fast and accurate.

---

[5]Except for DOM, where also a small contribution from the computation of optical properties is included

**Table 3.** Error statistics for all data sets (the ones from Tab. 1 and the ICON hindcasts for 12 UTC and 16 UTC), DOM computations for idealized profiles with different numbers of parameters (see Sect. 3.1) and three differently sized neural networks. '+BC' means that the bias correction from Sect. 3.5 was applied. Listed are the mean absolute error (MAE), the mean error (ME) or bias, and the 99th percentile of the absolute error (P99). For each metric three values are provided, which correspond to the albedo values $A = 0.0, 0.5$ and $1.0$. The ICON-D2 hindcasts are an exception – here we used for each profile only the actual albedo value.

| data | Setup | MAE | | | ME | | | P99 | | |
|---|---|---|---|---|---|---|---|---|---|---|
| | | A=0.0 | A=0.5 | A=1.0 | A=0.0 | A=0.5 | A=1.0 | A=0.0 | A=0.5 | A=1.0 |
| Standard | 1Lay, fix | 0.018 | 0.022 | 0.027 | 0.014 | 0.015 | 0.016 | 0.180 | 0.180 | 0.180 |
| | 1Lay | 0.017 | 0.017 | 0.018 | 0.015 | 0.014 | 0.014 | 0.180 | 0.180 | 0.180 |
| | 2Lay | 0.011 | 0.011 | 0.012 | 0.009 | 0.009 | 0.008 | 0.174 | 0.174 | 0.173 |
| | 2Lay+MP | 0.004 | 0.004 | 0.005 | 0.001 | 0.001 | 0.001 | 0.031 | 0.029 | 0.029 |
| | 2Lay+MP+BC | 0.003 | 0.003 | 0.004 | -0.001 | -0.001 | -0.002 | 0.026 | 0.024 | 0.023 |
| | NN2k | 0.007 | 0.009 | 0.012 | -0.003 | -0.004 | -0.005 | 0.046 | 0.042 | 0.042 |
| | NN5k | 0.004 | 0.005 | 0.007 | -0.002 | -0.001 | 0.001 | 0.029 | 0.027 | 0.029 |
| | NN8k | 0.004 | 0.005 | 0.007 | -0.002 | -0.003 | -0.003 | 0.028 | 0.026 | 0.029 |
| rmod | 1Lay, fix | 0.026 | 0.030 | 0.035 | 0.022 | 0.023 | 0.024 | 0.207 | 0.207 | 0.207 |
| | 1Lay | 0.025 | 0.025 | 0.026 | 0.023 | 0.023 | 0.022 | 0.205 | 0.205 | 0.205 |
| | 2Lay | 0.014 | 0.014 | 0.014 | 0.012 | 0.011 | 0.011 | 0.186 | 0.186 | 0.185 |
| | 2Lay+MP | 0.005 | 0.005 | 0.006 | 0.002 | 0.002 | 0.002 | 0.044 | 0.043 | 0.043 |
| | 2Lay+MP+BC | 0.004 | 0.004 | 0.005 | 0.000 | 0.000 | 0.000 | 0.037 | 0.035 | 0.035 |
| | NN2k | 0.008 | 0.011 | 0.013 | -0.001 | -0.002 | -0.003 | 0.050 | 0.047 | 0.048 |
| | NN5k | 0.005 | 0.006 | 0.008 | 0.000 | 0.001 | 0.003 | 0.038 | 0.036 | 0.037 |
| | NN8k | 0.005 | 0.006 | 0.008 | 0.000 | -0.001 | -0.002 | 0.039 | 0.037 | 0.038 |
| ICON-D2 12UTC | NN5k | | 0.010 | | | 0.002 | | | 0.046 | |
| ICON-D2 16UTC | NN5k | | 0.013 | | | 0.002 | | | 0.056 | |

## 5 Evaluation

In the previous sections, the reflectance errors related to simplifying vertical profiles and using neural networks instead of applying the DOM reference RT method to the simplified profiles have been discussed separately. We will now investigate the total error and the relative importance of simplification and neural network errors using also vertical profiles that are different from those used so far. To provide an overview over all considered cases, values for the mean absolute error (MAE), the mean error (ME) and the 99th percentile of the absolute error (P99), are listed in Tab. 3.

## 5.1 NWP SAF profiles

The mean absolute reflectance error for the 'std' profile data set with surface albedo $A = 0.5$ has already been shown in Fig. 10b as a function of the total optical depth and the maximum of the zenith angles, and the error was shown in Fig. 11b as a function of total optical depth. Here we investigate the full error distribution, discuss the relative impacts of the different input parameters and compare them to the effect the neural network errors have on the distribution. The reflectance error distributions of DOM computations with simplified profiles, $R_{1lay,fix}$, $R_{1lay}$, $R_{2lay}$, $R_{2lay,mp}$ and $R_{2lay,mp,bc}$, as well as neural network

calculations $R_{NN5k}$ are shown in Fig. 14 for the 'std' and 'rmod' data sets and different albedo values. In addition, values for the mean absolute error (MAE), the mean error (ME) and the 99th percentile of the absolute error (P99), are provided in Tab. 3.

It is quite obvious from the distribution plots and the metrics in Tab. 3 that DOM computations based on the 1Lay,fix idealised profiles (the original approach from Scheck et al. 2016) results in rather large errors, in particular on the left side of the histograms in Fig. 14 (light blue curves). Including surface pressure, cloud top pressure and integrated water vapour

improves the distribution significantly for positive reflectance errors, but not for the large negative errors (orange curves). Consequently, these improvements, which are related to profiles with high amounts of water vapour, results only in a reduction of the MAE, but not P99. The strongest MAE reductions are seen for $A = 1$, which can be expected, as in this case a wrong amount of water vapour leads to a largest error in the radiance reflected from the surface. Applying the two-layer approach (2Lay, purple curves) leads to further improvements for negative reflectance errors, resulting in lower mean and mean absolute

errors for all albedo values and both data sets. However, as many cases with large reflectance underestimation are still present, P99 still remains high. Improving the representation of mixed-phase clouds (2Lay+MP, red curves) basically removes all the extreme cases with negative error, reduces the P99s by about $80\%$ and the mean errors to very low levels. In 2Lay+MP, dark mixed-phase ice is often located below brighter water clouds, which leads to higher reflectances than in 2Lay, where all ice is always located above the water cloud. As already shown in Fig. 11, the remaining negative errors partly result from a optical

depth dependent bias. By applying the simple bias-correction from Sect. 3.5 the cases with larger errors are shifted to the right in the histograms (Fig. 14, green curves) without changing the position of the peak. Consequently, all metrics are slightly improved.

Comparing the computations with simplified profiles to results for the neural network NN5k (dark blue curves) shows that the additional network errors are small, compared to the simplification errors. In fact, the changes in histograms due to additional

NN errors are in most cases smaller than the ones caused by the bias correction. Only for $A = 1$ a somewhat stronger impact of the NN errors is visible. Results for the NN2k and NN8k networks (only included in Tab. 3, not in Fig. 14) show that the smaller 1983-parameter network causes non-negligible additional errors whereas the larger 8053-parameter network does not lead to significantly improved error metrics.

## 5.2 Regional ICON hindcasts

The results of the previous section show that our approach works also if effective radius parameterizations are used that are different from the ones employed for defining the two-layer splitting factor parameterizations and the bias correction. In the

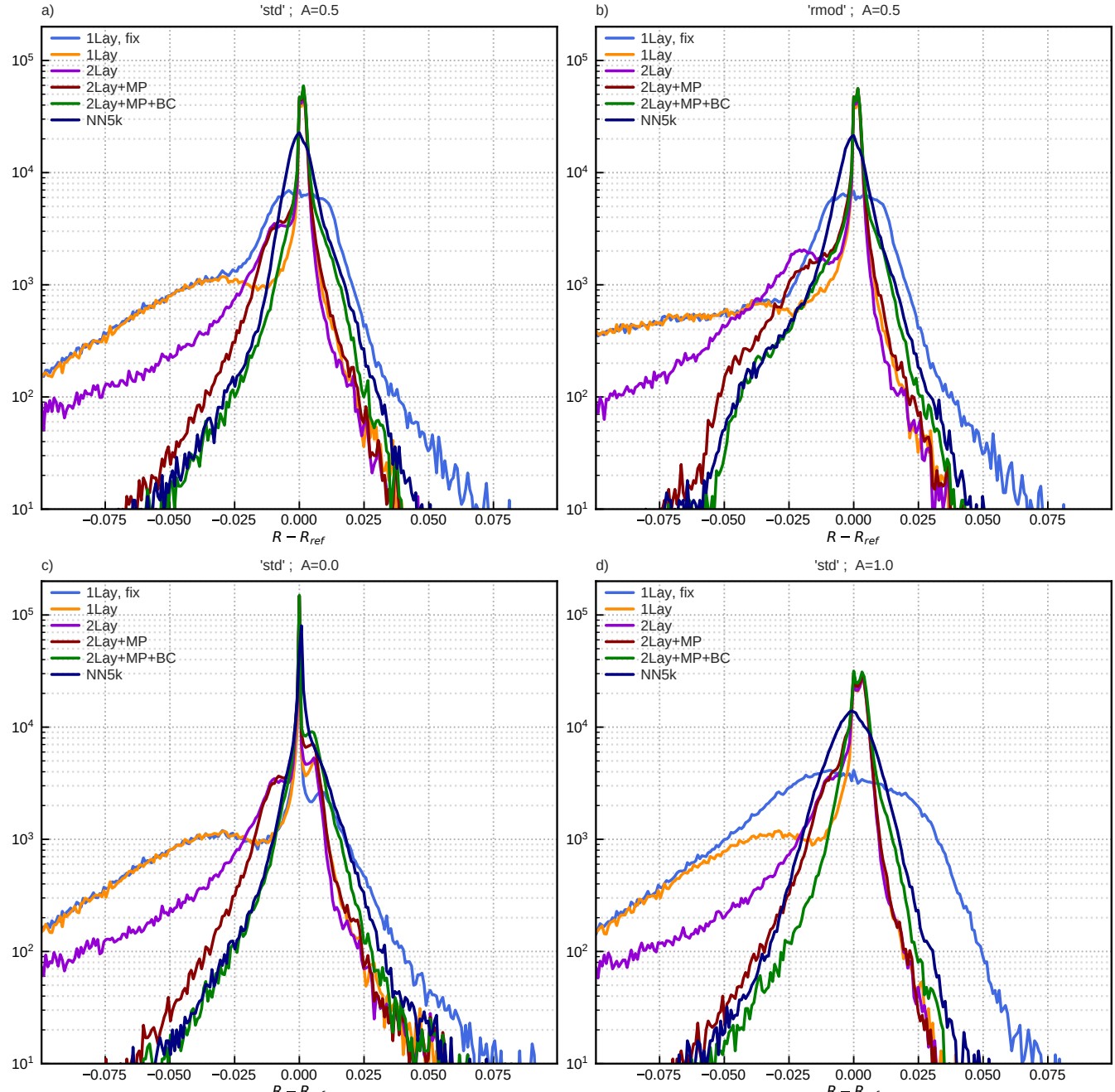

**Figure 14.** Error histogram showing the deviation of different reflectance estimations ($R$) from DOM reference computations ($R_{\text{ref}}$). Shown are estimations calculated using 1Lay,fix (light blue), 1Lay (orange), 2Lay (purple), 2-Layer parameterization adding mixed-phase clouds (2Lay+MP, red), 2-Layer parameterization adding mixed-phase clouds and bias-correction (2Lay+MP+BC, green), as well as the realization using the trained Neural Network (NN5k, dark blue). The different panels show the standard data (std) set applying albedo 0.5 (a), 0.0 (c) and 1.0 (d). Furthermore, rmod is shown for an albedo of 0.5 in panel b.

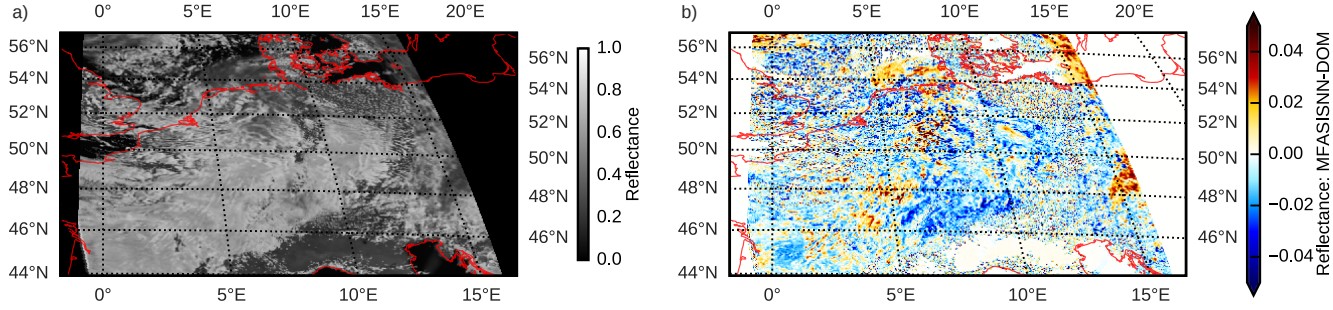

**Figure 15.** (a) Synthetic satellite image computed using the neural network NN5k for the ICON-D2 Domain on 4 June 2020 12UTC and (b) its deviation from the reference DOM computation.

following, we will investigate profiles from a different NWP model, the regional ICON-D2 model, and use effective radii determined by the two-moment microphysics scheme in the model.

We focus on synthetic SEVIRI $1.6\mu$m images of the ICON-D2 domain at 12UTC and 16UTC in a 30-day test period (June 2020). For each date and time, images were computed using both DOM and the NN5k network. In total, almost $14 \times 10^6$ test cases per time step are available for each time to compare DOM and the NN. As an example, Fig. 15a depicts the synthetic satellite image computed using NN5k for 4 June 2020 12UTC. Compared to the DOM reference method, the errors of NN5k are predominantly below 0.04 (Fig. 15b), which is in a similar range as for the 'std' and 'rmod' dat sets.

These results confirm the robustness of our approach. Although a completely different model and supposedly more realistic (and certainly different) effective radii are used, the errors are only slightly increased, compared to the ones computed for the IFS profile collection, which was used also for the development and optimization of the approach.

The error distribution for all 12 UTC and 16 UTC images of the test period (Fig. 16) is similar to the one for the 'rmod' IFS profile data set (compare to Fig. 14 panel b) blue line). The mean absolute error for the ICON-D2 case is only slightly higher MAE and P99 errors, but the overall mean error (ME) is similarly small (see also Tab. 3). The P99 for 16 UTC, when the solar zenith angle is larger, are only slightly worse.

## 5.3 Other solar channels

The method developed here for the $1.6\mu$m SEVIRI will not provide satisfactory results for all solar channels. However, for many visible and near-infrared channels the errors are in a similar range as for the $1.6\mu$m channel. To illustrate for which channels our method is usable, we consider all purely solar channels (we will not consider channels with thermal contributions like 3.7-3.9$\mu$m channels) of the current and next-generation EUMETSAT satellite imagers. These are SEVIRI on MSG, the Flexible Combined Imager (FCI) on Meteosat Third Generation (MTG), the Advanced Very High Resolution Radiometer (AVHRR) on the EUMETSAT Polar System (MetOp) and METimage on MetOp Second Generation. The root mean squared profile simplification error computed with the IFS profile collection for these channels is shown in Fig. 17. From these results it is obvious that all channels with wavelengths up to $0.7\mu$m and many channels with larger wavelengths up to $2.2\mu$m have errors

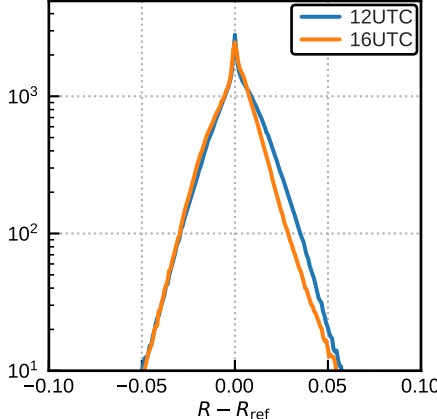

**Figure 16.** Error histogram showing the deviation of NN5k from the DOM reference computations. Data are sampled over the complete domain (see Fig. 15) for 12 UTC (blue) and 16UTC (orange). This data set amounts up to about $4.5 \times 10^6$ cloudy pixels.

similar to or smaller than the $1.6\mu$m channel. In particular, the stronger absorption by clouds in case of the $2.2\mu$m channel does not seem to be a problem. While stronger Rayleigh scattering could in principle pose an additional challenge for channels with small wavelengths, the error for these channels is actually smaller than for the $1.6\mu$m channels. It seems that the cloud top pressure and surface pressure input variables originally introduced to quantify the absorption by CO2 and CH4 characterize also the Rayleigh scattering sufficiently well. For those cases with a profile simplification RMSE smaller than 0.01 we trained

also neural networks and the full reflectance RMSE (due to profile simplification and imperfect network training) is shown as crosses in the Fig. 17. We did not train all networks for the same number of epochs and in some cases we used networks with only 2000 and not 5000 parameters, which results in some variation in the additional error related to the network training. In none of the cases the full reflectance RMSE exceeds 0.012 and further optimizations seem possible.

The channels for which the simplification error is rather large are the ones for which absorption by water vapour is stronger

than for the $1.6\mu$m SEVIRI channel and channels sensitive to certain absorption features like the $0.76\mu$m oxygen A band channel of MetOp SG. The influence of the spectral response function explains the difference between the $0.8\mu$m channels on FCI, MetImage, SEVIRI and AVHRR. On the older instruments the channels are wider and overlap more with water vapour absorption bands, whereas on the newer instruments the channels are narrower and experience less absorption. Quantifying the impact of water vapour with just one input parameter is sufficient for the newer instruments with their weak water vapour sen-

sitivity, but leads to larger errors for the more sensitive older ones. For all channels with higher simplification error, additional input parameters would be required to better quantify the influence of water vapour or other relevant gases.

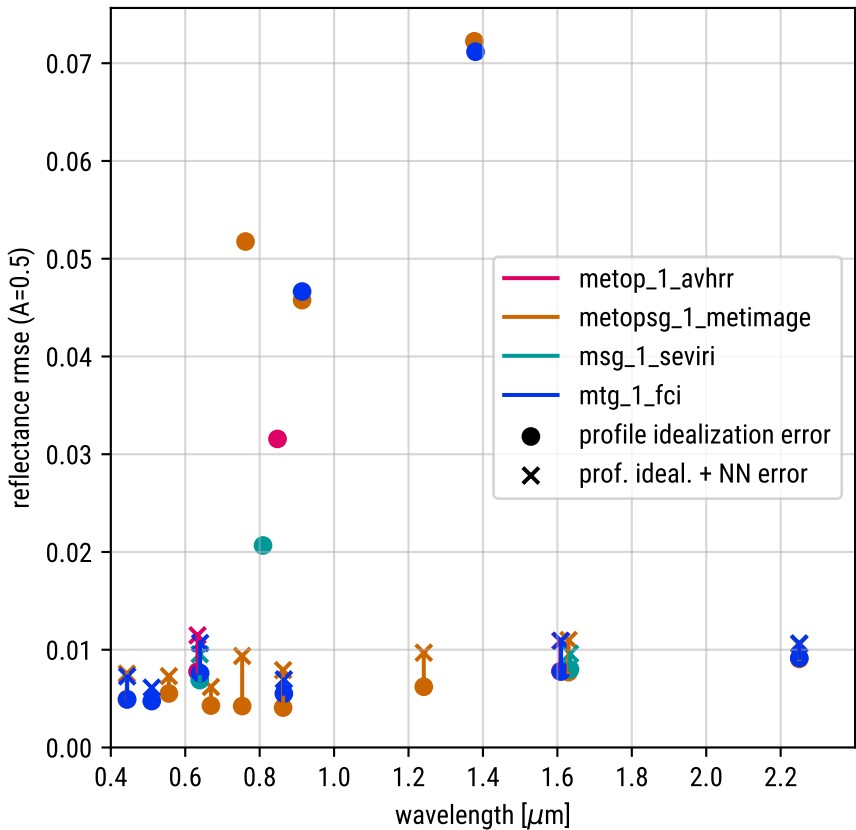

**Figure 17.** Root mean squared reflectance error due to profile simplification (circles) and, where available, full error (profile simplification and network training error, crosses) for all purely solar channels of the current and next generation of geostationary and polar orbiting imagers by EUMETSAT.

## 6 Conclusions

A computationally efficient, machine learning-based approach for the generation of synthetic $1.6\,\mu\text{m}$ near-infrared satellite images from NWP model output was developed. The new method is based on earlier work for visible channels, which involved a strong simplification of the cloud profiles from the NWP model and a feed forward neural network to predict reflectances from parameters defining the simplified profiles. For modelling the near-infrared channel, the representation of vertical effective radius gradients, mixed-phase clouds and molecular absorption was improved to achieve a similar accuracy as for the visible channel. The method was tested on a representative data set of IFS profiles using different effective radius parameterizations and additionally on profiles from the regional ICON-D2 model, which computes prognostic effective radii using a two-moment microphysics scheme. For all network sizes and test data sets, the mean absolute errors were found to be about an order of magnitude lower than typical observation errors assumed for the assimilation of visible channels, which are in the range 0.1-

0.15 for the case study of Scheck et al. (2020) and for the operational assimilation of SEVIRI $0.6\mu$m reflectances at DWD. The evaluation of the neural networks takes less than one microsecond per column. The method should therefore be suitably accurate and fast for operational data assimilation and model evaluation. For both applications the $1.6\mu$m channel provides valuable additional information that is complementary to the information content of visible and thermal infrared channels.

One of the next steps will be to perform assimilation experiments with both ensemble-based and variational data assimilation methods, which should include a discussion of the Jacobians of the neural network. Although the method was developed for $1.6\mu$m channels, it works also for many other solar channels, e.g. $2.2\mu$m channels and all channels with wavelengths up to $0.7\mu$m of imagers on the current and next-generation EUMETSAT satellites. For channels that are more sensitive to water vapour (like the $0.9\mu$m and $1.3\mu$m channels) or special absorption features (like $0.76\mu$m oxygen A band channels) the errors related to the profile simplification are still significantly higher, but could probably be reduced by means of additional input variables. The determination of suitable input parameters for the $1.6\mu$m channel required considerable effort. In a future study we will investigate whether the ability of neural networks to extract features can be used to automate parts of this process. The method presented in this study is aimed at generating synthetic images from NWP model profiles. It should be investigated whether such methods could also be applied on retrieved profiles, which could lead to further improvements and new applications. Finally, in this study we neglected 3D radiative effects, both in the design of the method and in our evaluation methods. Neglecting 3D effects can cause errors, in particular for high resolutions and large zenith angles. In future investigations we will test how well the approach of Scheck et al. (2018) to include 3D effects for visible channels works for near-infrared channels and whether it could be improved using machine learning (see also Zhou et al., 2021).

*Code and data availability.* FORNADO, the optimized Fortran inference code including tangent linear and adjoint versions used in this study is available from `https://gitlab.com/LeonhardScheck/fornado`. RTTOV including the DOM solver used for reference calculation can be obtained from `https://nwp-saf.eumetsat.int/site/software/rttov/rttov-v13`.
The IFS profile collection used for deriving and evaluating our methods is available from
`https://nwp-saf.eumetsat.int/site/software/atmospheric-profile-data`.

## Appendix A: Minimum and maximum integrated water vapour

The definition of a normalized integrated water vapour amount (Eq. 6) requires functions defining the minimum and maximum amount of water vapour above a given pressure level. Hard limits for the water vapour content for the pressure levels used in RTTOV are provided by the users guide (Hocking et al., 2020). For Eq. 6 we define the following smooth and differentiable functions that lie within the hard RTTOV limits, as visible in Fig. A1. The minimum amount of water vapour above a pressure level $p$ in units of millimeters is parameterized by

$$\mathrm{IWV}_{\min}(p) = 8.0 \times 10^{-11} \ f_{\min}(p) \ p^3 \ + \ 10^{-5} \left(1 - f_{\min}(p)\right) \ p \tag{A1}$$

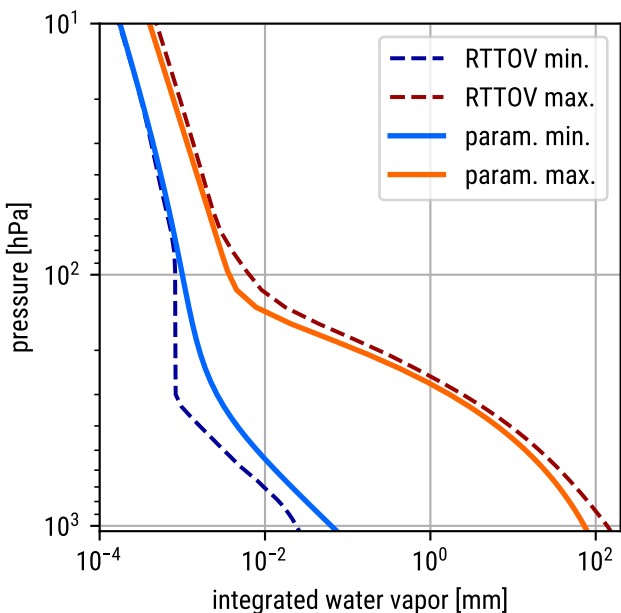

**Figure A1.** Minimum and maximum integrated water vapour amount above a given pressure. The dashed lines are integrals over the hard limits in Tab. 1 of Hocking et al. (2020), the solid lines represent the functions defined in Eqns. A1 and A2.

where $f_{\min}(p) = \tanh(\log(p) - \log(100))$, and the maximum amount by

$$\text{IWV}_{\max}(p) = 1.44 \times 10^3 \; f_{\max}(p) \; exp(\frac{-1.5 \times 10^3}{p}) \; + \; 4.14 \times 10^{-5} \; (1 - f_{\max}(p)) \; p \tag{A2}$$

with $f_{\max}(p) = \tanh(0.1 \; (\log(p) - \log(5)))$.

## 575   **Appendix B: Definition of idealised profiles**

The idealised profiles used to compute reflectances from the input parameters are based on the IFS 90-level standard atmosphere, which includes the geometric height $z_i^{\text{IFS}}$, the pressure $p_i^{\text{IFS}}$ and other variables for each level $i = 1, \ldots, 90$. To construct a grid starting at the desired surface pressure $p_{\text{sfc}}$, the corresponding height $z_{\text{sfc}}$ is obtained by interpolation in the standard atmosphere. Then a new vertical grid is defined as linear combination $\tilde{z}_i = z_i^{\text{IFS}} \times (1 - f_i) + (z_{\text{sfc}} + z_i^{\text{IFS}} - z_{\text{sfc}}^{\text{IFS}}) \times f_i$ of the

original grid and a grid shifted in the vertical such that the lowest level has the correct height. The factor $f_i = 1 - (z_i^{\text{IFS}}/z_{\text{sfc}}^{\text{IFS}})^2$ is chosen such that the original grid is retained for high altitudes and the shifted grid dominates at lower altitudes. For each level the pressure $\tilde{p}_i$ corresponding to $\tilde{z}_i$ is obtained by linear interpolation in the standard atmosphere. Finally, the level $i_{\text{ct}}$ in $\tilde{p}$ with a pressure closest to the desired cloud top pressure $p_{\text{ct}}$ is identified and the pressure on the five levels $i_{\text{ct}} - 2, \cdots, i_{\text{ct}} + 2$ is set to $[p_{\text{ct}} - \Delta p, p_{\text{ct}} - \Delta p/2, p_{\text{ct}}, p_{\text{ct}} + \Delta p/2, p_{\text{ct}} + \Delta p]$, where $\Delta p = \tilde{p}_{i_{\text{ct}}+1} - \tilde{p}_{i_{\text{ct}}}$ to obtain the final pressure grid $p_i$. Geo-

metric heights $z_i$ corresponding to these pressure levels are again computed by interpolation in the standard atmosphere. The two-layer ice cloud is placed in the two layers above the level $i_{ct}$ and the two-layer mixed-phase cloud in the two layers below.

*Author contributions.* Florian Baur and Leonhard Scheck designed and conducted experiments. Leonhard Scheck wrote the first draft with contributions from Florian Baur and Christina Köpken-Watts. Florian Baur and Leonhard Scheck produced the figures. All authors contributed to data interpretation and to revising the paper.

*Competing interests.* The autors declare that no competing interests are present.

*Acknowledgements.* This study was funded by the Hans-Ertel Centre for Weather Research. This German research network of universities, research institutes and DWD is funded by the BMVI (Federal Ministry of Transport and Digital Infrastructure, grant DWD2014P8). The work was also supported by the EUMETSAT Satellite Application Facility on Numerical Weather Prediction (NWP SAF). The Authors wish to thank Alberto de Lozar for providing the ICON-D2 Hindcasts. We would like to thank Hartwig Deneke and an anonymous reviewer for their detailed comments, which helped to improve this study significantly.

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
