# Peer review of "A neural network-based method for generating synthetic $1.6\text{ }\mu\text{m}$ near-infrared satellite images"

_EGUsphere, 2023_

## Referee Comment (RC2)

Review of the Manuscript "A neural network-based method for generating synthetic 1.6 μm near-infrared satellite images" Baur et al., doi:10.5194/egusphere-2023-353

The paper describes an extension of the existing MFASIS-NN satellite radiance forward operator

The paper is generally well-written, describes a novel scientific algorithm, and lies within the scope of AMT. As such, I recommend publication of the paper after addressing the following comments which summarizes my concerns about the present version of the manuscript.

General comments:

* I miss some discussion on the difficulties expected adapting this approach to other NIR/SWIR channels with cloud absorption, e.g. 2.2um or 3.7um, as well as the influence of the spectral response function (e.g. interaciton of gas absorption vs. droplet absorption). Is this just a matter of re-training the NN with different DOM forward simulations? If so, why has this not been done? Given that cloud particle absorption is comparatively weak at 1.6um vs. 2.2um, would this aspect influence the accuracy of the given approach? While I realize that fully covering this aspect would significantly expand the length of paper length, it seems worthwile to cover this point at least to some degree. At this stage, the focus on a single wavelength and instrument seems to unnecessarily limit the scope of the paper.

* Terminology: I have some reservations with the names used for the SEVIRI spectral channels. Frequently, the 1.6um and 2.2um channels are referred to as SWIR, and 0.8um is termed NIR[1]. While this might be a matter of taste, refering to 0.8um as VIS channel is misleading, as this wavelength is not within the range of human vision (even if this is the terminology used by EUMETSAT...).

* Performance: it would be good to give some more concrete indication of performance, beyond the two numbers given in the present manuscript. You state that "MFASIS-NN is an order of magnitude faster than MFASIS", and MFASIS is orders of magnitudes faster than running DOM. Maybe you can add a table of execution times of each algorithm in terms of pixels/profile calculations per second&CPU?

* The vertical variation of effective radius/ ice crystal size is purely based on parametrizations. What if these parametrizations are unrealistic? One could use A-Train profiles instead of IFS profiles to avoid this constrain. An alternative approach/extension could be to develop a set of representative basis profiles for different conditions / cloud types (e.g. similar to [1]). How well do these parametrizations capture the variability in effective particle size e.g. versus the ICON model hindcasts? I would really like to see this aspect/limitations discussed more in-depth, including possible ways improving this point in future research. Note that the treatment of vertical variations in cloud microphysics could also be used in cloud retrievals, giving guidance on selecting a target parameter set / limited number of degrees of freedom.

* Language: while the article is generally well-written some sentences would benefit from either being split or at least separating different aspects using a comma, and adding hyphens between words (e.g. L480, "machine learning based approach" => "machine learning-based approach").

Specific comments:

Abstract:
Given the paper content, I think the abstract can be clarified and improved to better describe the paper contents!
* L2: "with improved accuracy": the baseline for the "improved accuracy" should be clarified.
* L6: "vertical gradients": Gradients implies linearity, I therefore would prefer "vertical variations"
* L10: Sentence starting: "Additionally, a different parametrization … was used for testing". This sentence is suprising/unclea: please clarify explicitly the role of the "other" parametrization!
* L14: "in all cases, the mean absolute reflectance error achieved is about 0.01 or smaller". Is this with or without the "profile simplications" mentioned before? Can you add representative error estimates for the individual steps, e.g. going from DOM with fully known profiles to DOM with simplified profiles to MFASIS-NN?

Sec 1, Intro:
* L46: "An extension of MFASIS to account for the most important 3D effects….". Are these extensions applicable to the 1.6um capabilities presented in this paper? If not, what is the impact of 3D effects for the accuracy of the described method? In particular, it should be made clear that the chosen evaluation approach does not include an estimate of the resulting uncertainty.
* Paragraph starting at L49: I would recommend adding at least some context of the use of VIS channels plus the 1.6um channel in Nakajima-King style retrievals, and the fact that some of the challenges addressed in the present work are highly relevant for the resulting cloud products.
* L57: "because at this wavelength water clouds can be distinguished from ice clouds". I believe this statement is not true, there is an intermediate range were reflectances  (best reference I can find is this comment on a preprint in ACP, which raises concerns about the separability [2])

Sec2, Data and Methods
* L101: "they remain too large": How is "too large "determined? This statement implies an objective target accuracy, whose origin and magnitude should either be explicitly mentioned, or the statement should be reworded (e.g. "errors are significantly larger" ), to make it clear that this is a subjective statement.
* L102: "Sensitivity to the effective particle radii is higher": it remains unclear how sensitivity is defined here. Given the link between effective radius, optical depth and liquid water path, this statement only holds if optical depth is kept constant, not if liquid water path is kept constant!
* L111: role of water vapor absorption for SEVIRIs 1.6um channel could be described more clearly.
* L128: see Eq.2 in Scheck 2021. The equation reference seems to be wrong! The aspect of surface albedo also raises another interesting question: while this equation (referenced to Jonkerheid in Scheck 2021) can be used, why has the neural network not been trained to take surface albedo as input, and learn this equation?  Maybe the authors can comment on this?
* Sec2.3: I find the discussion if differences in effective particle size for water/ice clouds between ICON and the parametrizations too short and qualitative. What does e.g. "somewhat  smaller" mean?

Sec.3, Selecting Input parameters
* Figure 6: there seem to be a problem with the color bar, both online in Firefox and in my PDF viewer (okular), the color bar does not show a similar range of colors to the one visible in the figure! (Also applies to Figure 9 and 10!)

Sec.6 Conclusions
* L488: "that have been assumed in the assimilation": is there a reference for this number?

* L490: "the 1.6um provides": add channels

* L487: "in all cases, the mean … errors was about 0.01 or lower". Why is the number 0.01 given here? How does this number relate to the value given in L470 ("the errors of NN5k are predominantely below 0.04)? If I understand correctly, 0.01 refers to the comparison of DOM with simplified cloud profiles vs. the NN. For applications, isn't the larger number more relevant, which includes the error contribution resulting from the profile simplification?

[1] https://doi.org/10.5194/acp-23-2729-2023
[2] https://acp.copernicus.org/preprints/3/S1548/2003/acpd-3-S1548-2003.pdf

---

## Author Comment (AC1)

Response to Referee #1

In the following, our responses are in black, reviewer comments in *blue italics.*

*Review of the paper "A neural network-based method for generating synthetic 1.6 µm near-infrared satellite images" by Florian Baur et al., MS No.: egusphere-2023-353*

*This paper focuses on the development of a neural network to estimate the reflectance emerging from the atmosphere at 1.6 micron from SEVIRI on Meteosat Second Generation. This approach should also be suitable for other near-infrared channels on instruments such as AHI, ABI or FCI. This is a nicely written paper discusses the neural network performance when trained on reflectances calculated using DOM with both IFS and ICON-D input profiles.*

*I believe that this paper would be suitable for publication subject to minor revision addressing the specific comments detailed below.*

We are pleased to hear that and want to thank the reviewer for the constructive and helpful comments, which will try to address below.

*Specific comments*

*L37: here you should explain what you mean by 3D effects, e.g. reflection on complex topography etc. A reference could also be useful.*

We agree that an explanation should be added here, e.g. something like "By 3D effects we mean effects not taken into account in the plane-parallel RT equations, e.g. everything related to horizontal photon fluxes. One of the most important effects is the impact of inclined cloud tops on reflectance, for which a fast approximation was developed in Scheck et al. 2018. Other effects are related to cloud shadows, complex topography or photon transport through the cloud sides. A full discussion can be found in *Marshak & Davis (2005): 3D Radiative Transfer in Cloudy Atmospheres*".

*L50: here do you mean little information on discriminating the cloud phase?*

Yes. We will clarify that in the revised version.

*L66: It would be really interesting if you could show here an example of the water and ice cloud jacobians for the 1.6 micron channel, possibly by comparing to those for the visible (and/or thermal IR) channels.*

The derivatives with respect to the cloud variables depend strongly on the cloud structure. It is not obvious for which cloud profiles it would be interesting to see the Jacobians. Moreover, Jacobians are often complex and there are many different aspects that could be discussed. We think this information would be more relevant for a publication about the assimilation of near-infrared channels with a variational or hybrid assimilation method and we would thus prefer to postpone the discussion of Jacobians to a future study. As we plan also to add two more figures and several paragraphs of text in response to other comments of the two reviewers, we think the additional topic of the Jacobians would make the paper too long. What we would like to do, however, is to mention in the outlook that performing variational assimilation tests including a discussion of the Jacobians of the neural network should be one of the next steps.

*L84: How do you assess that the statement "are not very important" is true? Please add a reference to a paper where this is discussed and/or add a few explanations.*

We cannot find such a statement in line 84. You probably refer to the sentence "MFASIS makes use of the fact that for non-absorbing visible channels, the cloud top height and details of the vertical cloud structure are not very important." in lines 88/89. In MFASIS the cloud top height and details of the vertical cloud structure are not taken into account for computing the reflectance, and still the reflectance errors with respect to a reference solution are small. In this sense, these properties of the input profiles can be considered to be "not very important". We will clarify this in the revised version.

*L97: Do you mean interpolate the reflectance for the specific value of the albedo at a given location? If so you should expand your text here as you discuss this interpolation only later in the paper.*

Here we mean the linear interpolation in the optical depth, effective radii and angle dimensions of the LUT. In fact, interpolation in the albedo dimension is not necessary and we can exactly compute the reflectance as a function of the albedo, as we discuss later in the manuscript. We will clarify this in the revised version.

*L133: How many hidden layers, and did you test the effects of having more or fewer layers? how did you initialize the weights?*

We used 8 hidden layers and 15, 25 or 32 nodes per layer. We did some tests with fewer or more hidden layers (but roughly the same total number of parameters in the network) that did not show relevant differences. A more detailed discussion of this question can be found in Scheck 2021, where it was found that networks with between 4 and 8 hidden layers gave similar results. The weights were initialized using random numbers. We will add this information to the revised manuscript.

*L136: What distribution did you use for the random numbers? If uniform, which intervals?*

We used uniform random numbers between 0 and 1 for the normalized input parameters. The normalization was performed using the parameter ranges given by Table 2, as we will discuss more clearly in the revised version.

*L138: How many epochs were used in the training?*

We trained the networks for 4000 epochs, as specified in line 406. We can provide this information already in 2.1.3.

*Fig 2 caption: Please specify units of effective particle radius*

We will add the unit micrometers in the revised version.

*L234 ("exceeds a threshold value of 1"): Is this ok also for low optical depth clouds? From Fig 3 there are quite a few profiles with log(tau_w) ~ 0.1. Did you test having a threshold dependent on cloud optical depth categories? And did you test the radiative effects of the use of different thresholds?*

For low optical depth clouds the vertical cloud structure should not be very important any more, as reflectance is dominated by single-scattering and for this case it does not matter at which height the

scattering process takes place. We tested different values for the threshold and found that smaller threshold values did not significantly improve the results.

*L235 ("where z_sfc is the height…"): do you mean here the height of the highest model level below the bottom cloud layer?*

For the sake of simplicity, the lower bound $z^{l,bot}_w$ of the integral in Eq. 3 is set to the surface. We could provide a better definition for the bottom of the cloud layer, but this is just not necessary – if we integrate over cloud-free levels below the bottom cloud layer this will not change the integral. We will make this clear in the revised version.

*L244 ("exceeds a threshold value…"): I guess this at least partially answers my previous question. But I don't understand if this is done also for cth or only for ctp. And did you test other values (i.e. tau_t/5 etc.)*

The cloud top is defined as the level where the cloud optical depth exceeds the value given in the manuscript and this level defines also cloud top pressure and cloud top height.
The aim of the cloud top detection is to quantify the air mass above the cloud, as this quantity determines how strongly absorption by water vapor and trace gases influences the reflectance. We tested different threshold values. We found that the threshold proposed in the manuscript is a good compromise to prevent on the one hand that very thin, high clouds above thicker clouds trigger the cloud top detection and to avoid on the other hand that the cloud top is detected too deep inside the cloud.

*L255: It is not clear how these parameterizations are consistent with minimum and maximum values of IWP accepted by RTTOV*

We agree that this point was not discussed in sufficient detail. We will add the plot displayed below, which shows the minimum and maximum allowed IWV values above the pressure level given on the y-axis. The dashed lines are integrals over the hard humidity limits imposed by RTTOV (Table 1 in the RTTOV 13.2 user guide). The parameterizations (solid curves) lie between these limits. The focus was on using smooth, differentiable, not overly complicated functions rather than following the hard limits as closely as possible.

[Figure]

*L262: If I understand correctly, please replace "full vertical profiles" with "full idealised vertical profiles*

Correct, we will add "idealised".

*Fig 12 caption ("8 hidden layers with 25 nodes"): These numbers are inconsistent with those in the figure. Please correct the caption or the figure. Also, please check the total number of weights and biases is as stated.*

This is indeed inconsistent and will be changed in the revised version.

---

## Author Comment (AC2)

Response to Referee #2 (Hartwig Deneke)

In the following, our responses are in black, reviewer comments in *blue italics.*

*Review of the Manuscript "A neural network-based method for generating synthetic 1.6 µm near-infrared satellite images" Baur et al., doi:10.5194/egusphere-2023-353*

*The paper describes an extension of the existing MFASIS-NN satellite radiance forward operator*

*The paper is generally well-written, describes a novel scientific algorithm, and lies within the scope of AMT. As such, I recommend publication of the paper after addressing the following comments which summarizes my concerns about the present version of the manuscript.*

We are pleased to hear that and want to thank the reviewer for the constructive, detailed and helpful comments.

*General comments:*
*\* I miss some discussion on the difficulties expected adapting this approach to other NIR/SWIR channels with cloud absorption, e.g. 2.2um or 3.7um, as well as the influence of the spectral response function (e.g. interaciton of gas absorption vs. droplet absorption). Is this just a matter of re-training the NN with different DOM forward simulations? If so, why has this not been done? Given that cloud particle absorption is comparatively weak at 1.6um vs. 2.2um, would this aspect influence the accuracy of the given approach? While I realize that fully covering this aspect would significantly expand the length of paper length, it seems worthwile to cover this point at least to some degree. At this stage, the focus on a single wavelength and instrument seems to unnecessarily limit the scope of the paper.*

The methods developed here for the 1.6µm channel are not yet sufficient for all solar channels. We think it is justified to focus on this channel, as it provides important additional information, compared to visible channels, and is available on many existing satellite instruments. While we would like to keep the 1.6µm channel as the focus of our study, we fully agree with the reviewer that it would be useful to discuss for which other channels our methods are useful (without going too much into detail). For this purpose, we have evaluated our method for other instruments and channels.

As an example, the root mean squared profile simplification error for all purely solar channels (we will not consider channels with thermal contributions like 3.7-3.9µm channels) of the current and next-generation EUMETSAT satellite imagers is shown in the figure below (as circles). This figure gives a good indication on which channels are already usable with the current method. From the figure it is obvious that all channels with wavelengths up to 0.7µm and many channels with larger wavelengths up to 2.2µm have errors similar to or smaller than the 1.6µm channel. In particular, the stronger absorption by clouds in case of the 2.2µm does not seem to be a problem for our method. While stronger Rayleigh scattering could in principle pose an additional problem for channels with small wavelengths, the error for these channels is actually smaller than for the 1.6µm channels. It seems that the cloud top pressure and surface pressure input variables originally introduced to quantify the absorption by $CO_2$ and $CH_4$ characterize also the Rayleigh scattering sufficiently well. For those cases with a profile simplification RMSE smaller than 0.01 we trained also neural networks and the full reflectance RMSE (due to profile simplification and imperfect network training) is shown as crosses in the figure. We did not train all networks for the same number of epochs and in some cases we used networks with only 2000 and not 5000 parameters, which results in some variation in the additional error related to the network training. In none of the cases the full reflectance RMSE exceeds 0.012 and further optimizations seem possible.

The channels for which the simplification error is rather large are the ones for which absorption by water vapour is stronger than for the 1.6µm MSG channel and channels sensitive to certain trace gases like the 0.76µm oxygen A band channel of MetOp SG. The influence of the spectral response function explains the difference between the 0.8µm channels on FCI, MetImage, SEVIRI and AVHRR. On the older instruments the channels are wider and overlap more with water vapour absorption bands, whereas on the newer instruments the channels are more narrow and experience less absorption. Quantifying the impact of water vapour with just one input parameter is sufficient for the newer instruments with their weak water vapor sensitivity, but not for the more sensitive older ones. For all channels with higher simplification error, additional input parameters are required to better quantify the influence of water vapour or trace gases. In future studies we will investigate how to add suitable input parameters for these channels.

[Figure]

*Figure 1: Root mean squared reflectance error due to profile simplification (circles) and, where available, full error (profile simplification and network training error, crosses) for all purely solar channels of the current and next generation of geostationary and polar orbiting imagers by EUMETSAT.*

We will include the figure and the information given in this reply in the revised version.

*\* Terminology: I have some reservations with the names used for the SEVIRI spectral channels. Frequently, the 1.6um and 2.2um channels are referred to as SWIR, and 0.8um is termed NIR[1]. While this might be a matter of taste, refering to 0.8um as VIS channel is misleading, as this wavelength is not within the range of human vision (even if this is the terminology used by EUMETSAT...).*

We will adjust the terminology, avoiding to call 0.8µm "visible".

*\* Performance: it would be good to give some more concrete indication of performance, beyond the two numbers given in the present manuscript. You state that "MFASIS-NN is an order of magnitude faster than MFASIS", and MFASIS is orders of magnitudes faster than running DOM. Maybe you can add a table of execution times of each algorithm in terms of pixels/profile calculations per second&CPU?*

Exact measurements can already be found in Scheck 2021 (Fig. A12) for the 5000 parameter network and in Scheck et al. 2016 for the LUT-based method. We will include updated information in the manuscript.

*\* The vertical variation of effective radius/ ice crystal size is purely based on parametrizations. What if these parametrizations are unrealistic? One could use A-Train profiles instead of IFS profiles to avoid this constrain. An alternative approach/extension could be to develop a set of representative basis profiles for different conditions / cloud types (e.g. similar to [1]). How well do these parametrizations capture the variability in effective particle size e.g. versus the ICON model hindcasts? I would really like to see this aspect/limitations discussed more in-depth, including possible ways improving this point in future research. Note that the treatment of vertical variations in cloud microphysics could also be used in cloud retrievals, giving guidance on selecting a target parameter set / limited number of degrees of freedom.*

The method presented in our study is intended to speed up the computation of reflectance for cloud profiles from current global and regional NWP models. These models use either parameterizations or two-moment microphysics schemes for determining the effective radius profiles and so this is also what we are using for testing the method. As the method is primarily aimed at data assimilation and model evaluation we think this approach is sufficient for the current study.

While we agree that it would be interesting to check how well our method works for retrieved profiles, which could allow for additional applications, we think this is beyond the scope of this work and should be investigated in a separate, future study. Retrieved profiles may include features that are quite different from what is contained in model profiles (e.g. sharp gradients, extreme values) and these features and the distribution of integral quantities may not always be more realistic, but could also be related to limitations of the retrievals (see e.g. https://amt.copernicus.org/preprints/amt-2023-49/ ). Understanding the A-Train data set and selecting a suitable set of profiles could be a significant part of a future study, as well as approaches similar to the one in [1].

Concerning the question how well the parameterizations capture the variability of effective radii in the hindcasts, we agree that this should be discussed in more detail -- see response to the comment on Sect. 2.3 below.

We will add a more in-depth discussion on the aims and limitations of the method and future extensions. We will stress that the method is primarily aimed at data assimilation and model evaluation and discuss in the outlook that evaluation with retrieved profiles could be an interesting next step that may also lead to further improvements in out method.

*\* Language: while the article is generally well-written some sentences would benefit from either being split or at least separating different aspects using a comma, and adding hyphens between words (e.g. L480, "machine learning based approach" => "machine learning-based approach").*

We will try to address these issues.

*Specific comments:*

*Abstract:*
*Given the paper content, I think the abstract can be clarified and improved to better describe the paper contents!*

We agree that the abstract should be improved.

*\* L2: "with improved accuracy": the baseline for the "improved accuracy" should be clarified.*

We will clarify that we are aiming at an accuracy that is comparable to the one of existing operators for visible channels.

*\* L6: "vertical gradients": Gradients implies linearity, I therefore would prefer "vertical variations"*

We agree and will change that in a revised version.

*\* L10: Sentence starting: "Additionally, a different parametrization … was used for testing". This sentence is suprising/unclea: please clarify explicitely the role of the "other" parametrization!*

What was meant was that different parameterizations were used for testing the method. We will adjust the sentence.

*\* L14: "in all cases, the mean absolute reflectance error achieved is about 0.01 or smaller". Is this with or without the "profile simplications" mentioned before? Can you add representative error estimates for the individual steps, e.g. going from DOM with fully known profiles to DOM with simplified profiles to MFASIS-NN?*

The mean absolute error includes the error caused by profile simplification. The errors related to different profile simplification methods and the NN training are provided in Table 3 and discussed in detail in Section 5.1. We will adjust the abstract to make clear that the simplification error is dominating over the imperfect NN training error.

*Sec 1, Intro:*
*\* L46: "An extension of MFASIS to account for the most important 3D effects….". Are these extensions applicable to the 1.6um capabilities presented in this paper? If not, what is the impact of 3D effects for the accuracy of the described method? In particular, it should be made clear that the chosen evaluation approach does not include an estimate of the resulting uncertainty.*

In this study we just aim for replacing 1D radiative transfer methods like DOM by a faster alternative and 3D effects are not considered, as we will state more clearly in the manuscript. The 3D extension for MFASIS (Scheck et al. 2018) was developed for the 0.6µm channels and has not been tested for 1.6µm channel. In principle it should be possible to adjust the method for the 1.6µm channel, but this has to be investigated in a future study.

*\* Paragraph starting at L49: I would recommend adding at least some context of the use of VIS channels plus the 1.6um channel in Nakajima-King style retrievals, and the fact that some of the challenges addressed in the present work are highly relevant for the resulting cloud products.*

We will add a reference to the manuscript.

*L57: "because at this wavelength water clouds can be distinguished from ice clouds". I believe this statement is not true, there is an intermediate range were reflectances (best reference I can find is this comment on a preprint in ACP, which raises concerns about the separability [2])

We agree that this statement is not true for all cases (as visible e.g. in Fig. 4 of https://agupubs.onlinelibrary.wiley.com/doi/full/10.1029/2018JD029772 ). We will formulate more cautiously that the 1.6µm can provide information that is helpful for distinguishing water from ice clouds.

*Sec2, Data and Methods*
*L101: "they remain too large": How is "too large "determined? This statement implies an objective target accuracy, whose origin and magnitude should either be explicitly mentioned, or the statement should be reworded (e.g. "errors are significantly larger" ), to make it clear that this is a subjective statement.*

"Too large" is indeed not well-defined and "significantly larger" is a good replacement. What we meant is that the errors are not an order of magnitude smaller than the assumed observation error in VIS assimilation. For other applications like just producing an image for forecasters to look at the errors of the old method may already be acceptable.

*L102: "Sensitivity to the effective particle radii is higher": it remains unclear how sensitivity is defined here. Given the link between effective radius, optical depth and liquid water path, this statement only holds if optical depth is kept constant, not if liquid water path is kept constant!*

The reviewer is right -- it is only the derivative of reflectance with respect to the effective particle radius for constant optical depth that motivated this statement and we agree that the derivative for constant water/ice content is more relevant. While for water there is basically no difference between VIS006 and NIR016, the sensitivity to ice particle radii for constant ice content is actually somewhat larger for NIR016 (to give an example, if the radius is increase from 30µm to 60µm reflectance decreases by about 40% for VIS006 and 55% by NIR016).

We will reformulate the paragraph to make clear that the sensitivity to particle radius is not a decisive difference between VIS006 and NIR016. We will also change formulations in the introduction related to this question.

*L111: role of water vapor absorption for SEVIRIs 1.6um channel could be described more clearly.*

We will give a more quantitative description. Reflectance decreases approximately linearly with the water vapor mass. To give an example, for a relatively high column integrated water vapor content of 50mm and solar and satellite zenith angles of 60° the reduction is about 5%.

*L128: see Eq.2 in Scheck 2021. The equation reference seems to be wrong! The aspect of surface albedo also raises another interesting question: while this equation (referenced to Jonkerheid in Scheck 2021) can be used, why has the neural network not been trained to take surface albedo as input, and learn this equation? Maybe the authors can comment on this?*

It is indeed Eq. 6 that should have been referenced. If albedo was just used as an additional input parameter, the neural network would have to learn Eq. 6, and of course it would not be able to perfectly reproduce it. The errors related to this imperfect learning can be avoided by training the network to generate reflectances for three different albedo values and using Eq. 6. This has also the useful side effect that it is very cheap to compute reflectances for additional albedo values using Eq. 6. The need to do this arises e.g. in model columns containing land/sea or sea/sea-ice boundaries.

RTTOV14 will allow for specifying multiple albedo values (plus corresponding weights) and use Eq. 6 to compute the correct mean reflectance.

*\* Sec2.3: I find the discussion if differences in effective particle size for water/ice clouds between ICON and the parametrizations too short and qualitative. What does e.g. "somewhat smaller" mean?*

We agree that the discussion needs to be improved. We will add something like

"The effective radii from the two-moment scheme (Fig. 4) show qualitatively different dependences on the optical depths compared to those obtained with parameterizations (Fig. 2). For the two-moment calculations the mean effective droplet radius reaches a maximum at tau_w=100 and decreases for higher optical depths, whereas for the parameterization there a further increase for tau_w>100. The mean effective ice from the two-moment scheme are more similar to the Wyser (Fig. 2c) than to the McFarquhar (Fig. 2d) results, but show a minimum at tau=10, where for Wyser there is a minimum. Even more obvious differences between parameterized and two-moment radii are found for the spread. The parameterizations mostly depend on quantities strongly correlated with the optical depth, like LWC and IWC, and are therefore quite well-defined functions of the optical depth with a small spread. The only exception is the Wyser parameterization (Fig. 2d), which has an additional dependency on temperature and therefore a larger spread. In contrast, the two-moment radii always show a spread that is considerably larger than for the parameterizations for both water and ice clouds, as is to be expected for more realistic radii."

*Sec.3, Selecting Input parameters*
*\* Figure 6: there seem to be a problem with the color bar, both online in Firefox and in my PDF viewer (okular), the color bar does not show a similar range of colors to the one visible in the figure! (Also applies to Figure 9 and 10!)*

We will investigate this problem and correct the colors.

*Sec.6 Conclusions*
*\* L488: "that have been assumed in the assimilation": is there a reference for this number?*

We will cite Scheck et al. 2020 here.

*\* L490: "the 1.6um provides": add channels*

Ok.

*\* L487: "in all cases, the mean … errors was about 0.01 or lower". Why is the number 0.01 given here? How does this number relate to the value given in L470 ("the errors of NN5k are predominantely below 0.04)? If I understand correctly, 0.01 refers to the comparison of DOM with simplified cloud profiles vs. the NN. For applications, isn't the larger number more relevant, which includes the error contribution resulting from the profile simplification?*

Here we talk about the full mean absolut errors including simplification and NN training errors, i.e. the rows in Tab. 3 labeled NN2/5/8k. The point we want to make is that these errors are an order of magnitude smaller than the assumed observation errors in the VIS006 assimilation.

We would replace the sentence starting with "In all cases" by something like
"For all network sizes and test data sets the mean absolute errors were found to be about

an order of magnitude lower than typical observation errors assumed for the assimilation of visible channels, which are in the range 0.1-0.15 for Scheck et al. 2020 and also for the operational assimilation at DWD."

[1] https://doi.org/10.5194/acp-23-2729-2023
[2] https://acp.copernicus.org/preprints/3/S1548/2003/acpd-3-S1548-2003.pdf

---

## Author Response (AR1)

**Response to all comments on *egusphere-2023-353***

In the following, our responses are in black, changes to the manuscript are in red and reviewer comments in *blue italics.*

**Response to Referee #1**

*Review of the paper "A neural network-based method for generating synthetic 1.6 μm near-infrared satellite images" by Florian Baur et al., MS No.: egusphere-2023-353*

*This paper focuses on the development of a neural network to estimate the reflectance emerging from the atmosphere at 1.6 micron from SEVIRI on Meteosat Second Generation. This approach should also be suitable for other near-infrared channels on instruments such as AHI, ABI or FCI. This is a nicely written paper discusses the neural network performance when trained on reflectances calculated using DOM with both IFS and ICON-D input profiles.*

*I believe that this paper would be suitable for publication subject to minor revision addressing the specific comments detailed below.*

We are pleased to hear that and want to thank the reviewer for the constructive and helpful comments, which will try to address below.

*Specific comments*

*L37: here you should explain what you mean by 3D effects, e.g. reflection on complex topography etc. A reference could also be useful.*

We agree that an explanation is missing here and provide *Marshak & Davis (2005): 3D Radiative Transfer in Cloudy Atmospheres"* as reference.
An explanation including a reference was added (l. 39-41).

*L50: here do you mean little information on discriminating the cloud phase?*

Yes. We we have clarified that in the revised version.
Modified sentence (l. 53/54).

*L66: It would be really interesting if you could show here an example of the water and ice cloud jacobians for the 1.6 micron channel, possibly by comparing to those for the visible (and/or thermal IR) channels.*

The derivatives with respect to the cloud variables depend strongly on the cloud structure. It is not obvious for which cloud profiles it would be interesting to see the Jacobians. Moreover, Jacobians are often complex and there are many different aspects that could be discussed. We think this information would be more relevant for a publication about the assimilation of near-infrared channels with a variational or hybrid assimilation method and we would thus prefer to postpone the discussion of Jacobians to a future study. As we have added two more figures and several paragraphs of text in response to other comments of the two reviewers, we think the additional topic of the Jacobians would make the paper too long. However, in the revised version we mention now in the outlook that performing variational assimilation tests including a discussion of the Jacobians of the neural network should be one of the next steps.
Added assimilation experiments including discussion of Jacobians as next step in conclusions (l. 545/546).

*L84: How do you assess that the statement "are not very important" is true? Please add a reference to a paper where this is discussed and/or add a few explanations.*

We cannot find such a statement in line 84. You probably refer to the sentence "MFASIS makes use of the fact that for non-absorbing visible channels, the cloud top height and details of the vertical cloud structure are not very important." in lines 88/89. In MFASIS the cloud top height and details of the vertical cloud structure are not taken into account for computing the reflectance, and still the reflectance errors with respect to a reference solution are small. In this sense, these properties of the input profiles can be considered to be "not very important".
Clarified in the revised version (l. 95-97).

*L97: Do you mean interpolate the reflectance for the specific value of the albedo at a given location? If so you should expand your text here as you discuss this interpolation only later in the paper.*

Here we mean the linear interpolation in the optical depth, effective radii and angle dimensions of the LUT. In fact, interpolation in the albedo dimension is not necessary and we can exactly compute the reflectance as a function of the albedo, as we discuss later in the manuscript.
Added footnote about the interpolation dimensions to line 105

*L133: How many hidden layers, and did you test the effects of having more or fewer layers? how did you initialize the weights?*

We used 8 hidden layers and 15, 25 or 32 nodes per layer. We did some tests with fewer or more hidden layers (but roughly the same total number of parameters in the network) that did not show relevant differences. A more detailed discussion of this question can be found in Scheck 2021, where it was found that networks with between 4 and 8 hidden layers gave similar results. The weights were initialized using random numbers.
Added information on network structure and weight initialization (l. 143/144).

*L136: What distribution did you use for the random numbers? If uniform, which intervals?*

We used uniform random numbers between 0 and 1 for the normalized input parameters. The normalization was performed using the parameter ranges given by Table 2.
Added information on distribution (l. 148/149).

*L138: How many epochs were used in the training?*

We trained the networks for 4000 epochs, as specified in line 406. We provide this information now already in 2.1.3.
Number of epochs added (l. 150).

*Fig 2 caption: Please specify units of effective particle radius*

We added the unit micrometers in the revised version.

*L234 ("exceeds a threshold value of 1"): Is this ok also for low optical depth clouds? From Fig 3 there are quite a few profiles with log(tau_w) ~ 0.1. Did you test having a threshold dependent on cloud optical depth categories? And did you test the radiative effects of the use of different thresholds?*

For low optical depth clouds the vertical cloud structure should not be very important any more, as reflectance is dominated by single-scattering and for this case it does not matter at which height the scattering process takes place. We tested different values for the threshold and found that smaller threshold values did not significantly improve the results.
(No changes)

*L235 ("where z_sfc is the height…"): do you mean here the height of the highest model level below the bottom cloud layer?*

For the sake of simplicity, the lower bound $z^{l,bot}_w$ of the integral in Eq. 3 is set to the surface. We could provide a better definition for the bottom of the cloud layer, but this is just not necessary – if we integrate over cloud-free levels below the bottom cloud layer this will not change the integral.
Added explanation in footnote to line 264.

*L244 ("exceeds a threshold value…"): I guess this at least partially answers my previous question. But I don't understand if this is done also for cth or only for ctp. And did you test other values (i.e. tau_t/5 etc.)*

The cloud top is defined as the level where the cloud optical depth exceeds the value given in the manuscript and this level defines also cloud top pressure and cloud top height.
The aim of the cloud top detection is to quantify the air mass above the cloud, as this quantity determines how strongly absorption by water vapour and trace gases influences the reflectance. We tested different threshold values. We found that the threshold proposed in the manuscript is a good compromise to prevent on the one hand that very thin, high clouds above thicker clouds trigger the cloud top detection and to avoid on the other hand that the cloud top is detected too deep inside the cloud.
Added explanation in line 273-275.

*L255: It is not clear how these parameterizations are consistent with minimum and maximum values of IWP accepted by RTTOV*

We agree that this point was not discussed in sufficient detail. We added a plot displaying the minimum and maximum allowed IWV values above a given pressure level. Both the hard limits according to RTTOV and our parameterizations (that lie between the hard limits) are included. The focus was on using smooth, differentiable, not overly complicated functions rather than following the hard limits as closely as possible.
The new plot and the rather technical description of this point was moved to a new section in the appendix.

*L262: If I understand correctly, please replace "full vertical profiles" with "full idealised vertical profiles*

Correct, we added "idealised".
Done (now l. 287).

*Fig 12 caption ("8 hidden layers with 25 nodes"): These numbers are inconsistent with those in the figure. Please correct the caption or the figure. Also, please check the total number of weights and biases is as stated.*

This was indeed inconsistent and we changed it in the revised version.
Fig. 12 and caption corrected.

**Response to Referee #2 (Hartwig Deneke)**

*Review of the Manuscript "A neural network-based method for generating synthetic 1.6 μm near-infrared satellite images" Baur et al., doi:10.5194/egusphere-2023-353*

*The paper describes an extension of the existing MFASIS-NN satellite radiance forward operator*

*The paper is generally well-written, describes a novel scientific algorithm, and lies within the scope of AMT. As such, I recommend publication of the paper after addressing the following comments which summarizes my concerns about the present version of the manuscript.*

We are pleased to hear that and want to thank the reviewer for the constructive, detailed and helpful comments.

*General comments:*
*\* I miss some discussion on the difficulties expected adapting this approach to other NIR/SWIR channels with cloud absorption, e.g. 2.2um or 3.7um, as well as the influence of the spectral response function (e.g. interaciton of gas absorption vs. droplet absorption). Is this just a matter of re-training the NN with different DOM forward simulations? If so, why has this not been done? Given that cloud particle absorption is comparatively weak at 1.6um vs. 2.2um, would this aspect influence the accuracy of the given approach? While I realize that fully covering this aspect would significantly expand the length of paper length, it seems worthwile to cover this point at least to some degree. At this stage, the focus on a single wavelength and instrument seems to unnecessarily limit the scope of the paper.*

The methods developed here for the 1.6µm channel are not yet sufficient for all solar channels. We think it is justified to focus on this channel, as it provides important additional information, compared to visible channels, and is available on many existing satellite instruments. While we would like to keep the 1.6µm channel as the focus of our study, we fully agree with the reviewer that it would be useful to discuss for which other channels our methods are useful (without going too much into detail). For this purpose, we have evaluated our method for other instruments and channels using the IFS profile collection. As an example, we discuss now all purely solar channels (we will not consider channels with thermal contributions like 3.7-3.9µm channels) of the current and next-generation EUMETSAT satellite imagers in the revised version. Our results show that all channels with wavelengths up to 0.7µm and many channels with larger wavelengths up to 2.2µm have errors similar to or smaller than the 1.6µm channel. In particular, the stronger absorption by clouds in case of the 2.2µm does not seem to be a problem for our method. We discuss also the impact of Rayleigh scattering and water vapour absorption as well as the training of the networks for these channels.
We added a subsection "5.3 Other solar channels" to the results section, including a plot with simplification errors and total errors.

*\* Terminology: I have some reservations with the names used for the SEVIRI spectral channels. Frequently, the 1.6um and 2.2um channels are referred to as SWIR, and 0.8um is termed NIR[1]. While this might be a matter of taste, refering to 0.8um as VIS channel is misleading, as this wavelength is not within the range of human vision (even if this is the terminology used by EUMETSAT...).*

We will adjust the terminology, avoiding to call 0.8µm "visible".
Done.

*\* Performance: it would be good to give some more concrete indication of performance, beyond the two numbers given in the present manuscript. You state that "MFASIS-NN is an order of magnitude faster than MFASIS", and MFASIS is orders of magnitudes faster than running DOM. Maybe you can add a table of execution times of each algorithm in terms of pixels/profile calculations per second&CPU?*

We have now included benchmarks for the neural network, the LUT-based MFASIS and DOM on a current server CPU.

Added benchmarks to in 4.2 (l. 441-445).

*\* The vertical variation of effective radius/ ice crystal size is purely based on parametrizations. What if these parametrizations are unrealistic? One could use A-Train profiles instead of IFS profiles to avoid this constrain. An alternative approach/extension could be to develop a set of representative basis profiles for different conditions / cloud types (e.g. similar to [1]). How well do these parametrizations capture the variability in effective particle size e.g. versus the ICON model hindcasts? I would really like to see this aspect/limitations discussed more in-depth, including possible ways improving this point in future research. Note that the treatment of vertical variations in cloud microphysics could also be used in cloud retrievals, giving guidance on selecting a target parameter set / limited number of degrees of freedom.*

The method presented in our study is intended to speed up the computation of reflectance for cloud profiles from current global and regional NWP models. These models use either parameterizations or two-moment microphysics schemes for determining the effective radius profiles and so this is also what we are using for testing the method. As the method is primarily aimed at data assimilation and model evaluation we think this approach is sufficient for the current study.

While we agree that it would be interesting to check how well our method works for retrieved profiles, which could allow for additional applications, we think this is beyond the scope of this work and should be investigated in a separate, future study. Retrieved profiles may include features that are quite different from what is contained in model profiles (e.g. sharp gradients, extreme values) and these features and the distribution of integral quantities may not always be more realistic, but could also be related to limitations of the retrievals (see e.g. https://amt.copernicus.org/preprints/amt-2023-49/ ). Understanding the A-Train data set and selecting a suitable set of profiles could be a significant part of a future study, as well as approaches similar to the one in [1].

Concerning the question how well the parameterizations capture the variability of effective radii in the hindcasts, we agree that this should be discussed in more detail --  see response to the comment on Sect. 2.3 below.

We have added a more in-depth discussion on the aims and limitations of the method and future extensions. We will stress that the method is primarily aimed at data assimilation and model evaluation and discuss in the outlook that evaluation with retrieved profiles could be an interesting next step that may also lead to further improvements in out method.

Changes in the paper: New paragraph at the end of 2.2, improved discussion at the end of 2.3, changes in introduction (l. 75) and conclusions (l. 553-555).

*\* Language: while the article is generally well-written some sentences would benefit from either being split or at least separating different aspects using a comma, and adding hyphens between words (e.g. L480, "machine learning based approach" => "machine learning-based approach").*

We will try to address these issues.

We made various small changes throughout the paper.

*Specific comments:*

*Abstract:*
*Given the paper content, I think the abstract can be clarified and improved to better describe the paper contents!*

We agree that the abstract should be improved.

Abstract rewritten.

*\* L2: "with improved accuracy": the baseline for the "improved accuracy" should be clarified.*

We will clarify that we are aiming at an accuracy that is comparable to the one of existing operators for visible channels.

Done (l. 6).

*\* L6: "vertical gradients": Gradients implies linearity, I therefore would prefer "vertical variations"*

Done (l. 7).

*\* L10: Sentence starting: "Additionally, a different parametrization … was used for testing". This sentence is suprising/unclea: please clarify explicitely the role of the "other" parametrization!*

What was meant was that different parameterizations were used for testing the method.

Clarified (l. 10).

*\* L14: "in all cases, the mean absolute reflectance error achieved is about 0.01 or smaller". Is this with or without the "profile simplications" mentioned before? Can you add representative error estimates for the individual steps, e.g. going from DOM with fully known profiles to DOM with simplified profiles to MFASIS-NN?*

The mean absolute error includes the error caused by profile simplification. The errors related to different profile simplification methods and the NN training are provided in Table 3 and discussed in detail in Section 5.1. We adjusted the abstract to make clear that the simplification error is dominating over the imperfect NN training error.

Abstract adjusted (l. 14/15).

*Sec 1, Intro:*
*\* L46: "An extension of MFASIS to account for the most important 3D effects….". Are these extensions applicable to the 1.6um capabilities presented in this paper? If not, what is the impact of 3D effects for the accuracy of the described method? In particular, it should be made clear that the chosen evaluation approach does not include an estimate of the resulting uncertainty.*

In this study we just aim for replacing 1D radiative transfer methods like DOM by a faster alternative and 3D effects are not considered, as is now stated more clearly in the manuscript. The 3D extension for MFASIS (Scheck et al. 2018) was developed for the 0.6µm channels and has not been tested for 1.6µm channel. In principle it should be possible to use or adapt the method for the 1.6µm channel, but this has to be investigated in a future study.

Added discussion in conclusions (l. 556-559).

*\* Paragraph starting at L49: I would recommend adding at least some context of the use of VIS channels plus the 1.6um channel in Nakajima-King style retrievals, and the fact that some of the challenges addressed in the present work are highly relevant for the resulting cloud products.*

We added the information that VIS+NIR has been used for many years to simultaneously retrieve cloud optical thickness and effective particle radii following Nakajima and King (1990).
Added sentence with reference (l. 58/59).

*\* L57: "because at this wavelength water clouds can be distinguished from ice clouds". I believe this statement is not true, there is an intermediate range were reflectances (best reference I can find is this comment on a preprint in ACP, which raises concerns about the separability [2])*

We agree that this statement is not true for all cases (as visible e.g. in Fig. 4 of https://agupubs.onlinelibrary.wiley.com/doi/full/10.1029/2018JD029772 ). We formulated now more cautiously that the 1.6µm can provide information that is helpful for distinguishing water from ice clouds.
Changed formulation, added reference (l. 63-65).

*Sec2, Data and Methods*
*\* L101: "they remain too large": How is "too large "determined? This statement implies an objective target accuracy, whose origin and magnitude should either be explicitly mentioned, or the statement should be reworded (e.g. "errors are significantly larger" ), to make it clear that this is a subjective statement.*

"Too large" is indeed not well-defined and "significantly larger" is a good replacement. What we meant is that the errors are not an order of magnitude smaller than the assumed observation error in VIS assimilation. For other applications like just producing an image for forecasters to look at the errors of the old method may already be acceptable.
Changed to "significantly larger", specified data assimilation of model evaluation as intended applications (l. 108/109).

*\* L102: "Sensitivity to the effective particle radii is higher": it remains unclear how sensitivity is defined here. Given the link between effective radius, optical depth and liquid water path, this statement only holds if optical depth is kept constant, not if liquid water path is kept constant!*

The reviewer is right – it is only the derivative of reflectance with respect to the effective particle radius for constant optical depth that motivated this statement and we agree that the derivative for constant water/ice content is more relevant here. While for water there is basically no difference between VIS006 and NIR016, the sensitivity to ice particle radii for constant ice content is actually somewhat larger for NIR016 (to give an example, if the radius is increase from 30µm to 60µm reflectance decreases by about 40% for VIS006 and 55% by NIR016).

We reformulated the paragraph to make clear that the sensitivity to particle radius is not a decisive difference between VIS006 and NIR016 and made also changes to the introduction to take this into account.
New formulations and reordered list in 2.1.2, removed "Near-infrared channels are more sensitive to cloud droplet and ice particle sizes" from introduction.

*L111: role of water vapor absorption for SEVIRIs 1.6um channel could be described more clearly.*

We will give a more quantitative description. Reflectance decreases approximately linearly with the water vapour mass. To give an example, for a relatively high column integrated water vapour content of 50mm and solar and satellite zenith angles of 60° the reduction is about 5%.
Added information on water vapour impact (l. 116/117).

*\* L128: see Eq.2 in Scheck 2021. The equation reference seems to be wrong! The aspect of surface albedo also raises another interesting question: while this equation (referenced to Jonkerheid in Scheck 2021) can be used, why has the neural network not been trained to take surface albedo as input, and learn this equation? Maybe the authors can comment on this?*

It is indeed Eq. 6 that should have been referenced. If albedo was just used as an additional input parameter, the neural network would have to learn Eq. 6, and of course it would not be able to perfectly reproduce it. The errors related to this imperfect learning can be avoided by training the network to generate reflectances for three different albedo values and using Eq. 6. This has also the useful side effect that it is very cheap to compute reflectances for additional albedo values using Eq. 6. The need to do this arises e.g. in model columns containing land/sea or sea/sea-ice boundaries. RTTOV14 will allow for specifying multiple albedo values (plus corresponding weights) and use Eq. 6 to compute the correct mean reflectance.
Corrected reference (l. 138).

*\* Sec2.3: I find the discussion if differences in effective particle size for water/ice clouds between ICON and the parametrizations too short and qualitative. What does e.g. "somewhat smaller" mean?*

We agree that the discussion needed to be improved. We discuss now Fig. 4 in more detail. We also found and removed an error in the script generating Fig. 4b, which has led to larger radii for higher optical depths.
Added discussion at the end of 2.3, removed "somewhat smaller", corrected Fig. 4.

*Sec.3, Selecting Input parameters*
*\* Figure 6: there seem to be a problem with the color bar, both online in Firefox and in my PDF viewer (okular), the color bar does not show a similar range of colors to the one visible in the figure! (Also applies to Figure 9 and 10!)*

The colour bar problem should be solved now.
Repaired figures 6, 9, 10.

*Sec.6 Conclusions*
*\* L488: "that have been assumed in the assimilation": is there a reference for this number?*

Scheck et al. 2020 and the operational setup at DWD are now given as references.
Done (l. 542).

*\* L490: "the 1.6um provides": add channels*

Done (l. 544).

*\* L487: "in all cases, the mean … errors was about 0.01 or lower". Why is the number 0.01 given here? How does this number relate to the value given in L470 ("the errors of NN5k are predominantely below 0.04)? If I understand correctly, 0.01 refers to the comparison of DOM with*

*simplified cloud profiles vs. the NN. For applications, isn't the larger number more relevant, which includes the error contribution resulting from the profile simplification?*

Here we talk about the full mean absolut errors including simplification and NN training errors, i.e. the rows in Tab. 3 labeled NN2/5/8k. The point we want to make is that these errors are an order of magnitude smaller than the assumed observation errors in the VIS006 assimilation. We clarified the sentence starting with "In all cases".
Changed sentence in conclusions (l. 540-542).

[1] https://doi.org/10.5194/acp-23-2729-2023
[2] https://acp.copernicus.org/preprints/3/S1548/2003/acpd-3-S1548-2003.pdf